# PEX3 promotes regenerative repair after myocardial injury in mice through facilitating plasma membrane localization of ITGB3
Jia-Teng Sun[1,6], Zi-Mu Wang[1,6], Liu-Hua Zhou[1,6], Tong-Tong Yang[1,6], Di Zhao[1], Yu-Lin Bao[1], Si-Bo Wang [1], Ling-Feng Gu [1], Jia-Wen Chen[1], Tian-Kai Shan[1], Tian-Wen Wei[1], Hao Wang[1], Qi-Ming Wang[1], Xiang-Qing Kong[1], Li-Ping Xie[2], Ai-Hua Gu[3], Yang Zhao[4], Feng Chen[4], Yong Ji [2], Yi-Qiang Cui[5] ✉ & Lian-Sheng Wang [1] ✉

The peroxisome is a versatile organelle that performs diverse metabolic functions. PEX3, a critical regulator of the peroxisome, participates in various biological processes associated with the peroxisome. Whether PEX3 is involved in peroxisome-related redox homeostasis and myocardial regenerative repair remains elusive. We investigate that cardiomyocyte-specific PEX3 knockout (Pex3-KO) results in an imbalance of redox homeostasis and disrupts the endogenous proliferation/ development at different times and spatial locations. Using Pex3-KO mice and myocardium-targeted intervention approaches, the effects of PEX3 on myocardial regenerative repair during both physiological and pathological stages are explored. Mechanistically, lipid metabolomics reveals that PEX3 promotes myocardial regenerative repair by affecting plasmalogen metabolism. Further, we find that PEX3-regulated plasmalogen activates the AKT/GSK3β signaling pathway via the plasma membrane localization of ITGB3. Our study indicates that PEX3 may represent a novel therapeutic target for myocardial regenerative repair following injury.

Ischemic heart disease is the leading cause of morbidity and mortality worldwide[1]. In recent years, efforts to identify the endogenous mammalian myocardial regeneration window have opened up new possibilities in cardiac repair[2]. Neonatal mice have the remarkable ability to undergo complete structural and functional recovery through endogenous myocardial regeneration following myocardial infarction (MI) or apical resection (AR). However, this regenerative capacity gradually diminishes after birth[3]. Previous research has shown that mammalian cardiomyocytes (CMs) undergo a metabolic shift from anaerobic glycolysis to mitochondrial oxidative phosphorylation during the transition from a low-oxygen state in utero to a high-oxygen environment after birth[4]. However, this metabolic transition leads to the accumulation of reactive oxygen species (ROS) as by-products, which can cause oxidative damage to DNA and lead to the loss of the regenerative capacity of CMs[5]. Studies demonstrated that timely and efficient elimination of ROS was crucial for maintaining and prolonging the myocardial proliferation and regenerative ability[6,7]. Lalowski et al. found that besides ROS formation, oxidative stress changes (such as lipid peroxidation, sphingolipid and plasmalogen metabolism) are also responsible for the loss of regenerative capacity after birth[8].

The peroxisome is composed of a membrane enclosing a finely granular matrix. Peroxisome is known to play a critical role in lipid metabolism and maintaining the homeostasis of ROS[9]. A group of heterogeneous

[1]Department of Cardiology, The First Affiliated Hospital of Nanjing Medical University, Nanjing 210029, China. [2]Key Laboratory of Cardiovascular and Cerebrovascular Medicine, Key Laboratory of Targeted Intervention of Cardiovascular Disease, Collaborative Innovation Center for Cardiovascular Disease Translational Medicine, Nanjing Medical University, Nanjing, China. [3]State Key Laboratory of Reproductive Medicine, School of Public Health, Nanjing Medical University, Nanjing, China. [4]Department of Biostatistics, School of Public Health, China International Cooperation Center for Environment and Human Health, Nanjing Medical University, Nanjing 210029, China. [5]State Key Laboratory of Reproductive Medicine, Department of Histology and Embryology, Nanjing Medical University, Nanjing 210029, China. [6]These authors contributed equally: Jia-Teng Sun, Zi-Mu Wang, Liu-Hua Zhou, Tong-Tong Yang. ✉e-mail: cuiyiqiang@126.com; drlswang@njmu.edu.cn

proteins called peroxins (PEXs) are responsible for the occurrence of peroxisome biological processes and regulating multiple functions such as proliferation and maturation[10]. The significance of peroxisome in cardiovascular disease is increasingly recognized. Fahimi et al. discovered that inhibition of peroxisome activity led to myocardial vacuolation, focal necrosis and fibrosis. Besides maintaining redox homeostasis, the biosynthesis of plasmalogens also starts from peroxisomes[11,12].

As a crucial regulatory factor of peroxisome, peroxisome biogenesis factor 3 (PEX3) exerts dynamic effects on the peroxisomal membrane. PEX3, interacting with various proteins, is known to participate in peroxisome biosynthesis and kinetic modulation[10,13]. Several studies have indicated that PEX3 is involved in the processes related to direct or indirect targeting and sorting of peroxisome membrane proteins, thereby facilitating peroxisome maturation and turnover. In addition, many studies have found that PEX3 is also helpful in attenuating oxidative stress-mediated DNA damage in various systemic diseases[14–16]. In this study, we tested the hypothesis that PEX3 might play a crucial role in myocardial repair by modulating peroxisome metabolism.

## Results

### PEX3-mediated peroxisome function is associated with regenerative repair after myocardial injury

Following myocardial injury, the newborn mouse demonstrated impressive regeneration and functional restoration capability. In contrast, adult mice tend to undergo scar formation within the injured region as part of the healing process. To examine the potential involvement of peroxisome in myocardial injury repair, AR models in neonatal mice and MI models in adult mice were created to observe whether there is a differential expression of peroxisome between these two distinct models of myocardial repair. Immunofluorescence (IF) results showed that PMP70 (a peroxisome marker) was mainly activated in CMs after neonatal myocardial injury, while decreased in CMs after adult myocardial injury (Supplemental Fig. 1a, b). Western blot (WB) analysis demonstrated a significant increase in the expression of the PMP70 following neonatal myocardial injury, whereas there was a decrease in PMP70 expression after adult myocardial injury (Supplemental Fig. 1c, d). Immunohistochemistry (IHC) results showed a substantial increase of PMP70 in the injury zone and border zone after neonatal myocardial injury compared with that in the normal zone, while PMP70 expression decreased in the injury zone and border zone after adult myocardial injury compared with that in the normal zone (Supplemental Fig. 1e).

Peroxisome is regulated and formed by a group of heterogeneous protein families. Therefore, we further assessed the heterogeneous changes in PEXs after neonatal myocardial injury. The main biological functions of each member of the PEX family are summarized in Supplemental Table 1. The qRT-PCR results revealed an upregulation of peroxisome-related genes during the injury-activated regeneration process in the neonatal myocardium, with the most notable change observed in PEX3 (Fig. 1a). Subsequently, WB analysis demonstrated that the expression of PEX3 and PEX12 increased after neonatal myocardial injury, while the increase of PEX3 was more significant (Fig. 1b). And we found that the expression of PEX3 significantly decreases after adult myocardial injury (Fig. 1c). Bulk RNA-seq data from mice hearts at various developmental stages indicated that Pex3 exhibited high expression levels at postnatal day 1 (P1), followed by a significant decrease in expression after P7[17] (Fig. 1d). Further, IF and IHC assays demonstrated that PEX3 was mainly activated in CMs after neonatal myocardial injury, whereas a significant decrease in PEX3 expression was observed in CMs after adult myocardial injury, similar to the same expression pattern of PMP70 (Supplemental Fig. 2a–c, Fig. 1e, f). Additionally, PEX3 expression was significantly decreased after ischemic injury in myocardium samples of adult humans[18] (Supplemental Fig. 2d). The above results suggested that the increased peroxisome functions and its regulator PEX3 were significantly correlated with the activation of myocardial regenerative repair.

### Blocked endogenous myocardial proliferation and development block in CM-specific PEX3 knockout mice

To investigate the role of PEX3 in myocardial regenerative repair, we constructed CM-specific Pex3 knockout (Pex3-KO) mice by breeding Myh6-Cre mice with PEX3 floxed mice. (Supplemental Fig. 3a). Also, qRT-PCR and WB analysis were used to demonstrate that Pex3-KO was successful in CMs (Supplemental Fig. 3b–d). Based on our preliminary findings, which suggested a significant temporal and spatial correlation between PEX3 and myocardial regenerative ability, we investigated the impact of Pex3-KO on the proliferation and development of normal myocardium (Fig. 2a). Ki67 (a cell cycle activity index) and pH3 (a mitotic index) assays were used to assess the endogenous proliferation of CMs from P1 to P28. The results indicated that the endogenous myocardial proliferation capacity of Pex3-KO neonatal mice was significantly reduced compared to that of Wild type (WT) mice at P7. Furthermore, the myocardial proliferation capacity of Pex3-KO mice remained markedly lower than that of WT mice in the following stage (Fig. 2b, c). As expected, Pex3-KO hearts showed a low percentage of mononucleated CMs starting from P3 (Fig. 2d). We measured the heart weight and body weight in Pex3-KO and WT mice at different times (P1, P3, P7, P14, P28). During the development of the Pex3-KO mice, both heart weight and body weight decreased significantly, but there was no significant difference in the heart weight ratio (heart weight/body weight) (Supplemental Fig. 3e, f, Fig. 2e).

At P28, Pex3-KO mice presented with cardiac morphology dysplasia, but wheat germ agglutinin (WGA) staining subsequently showed no significant difference in the size of CMs compared with that in WT hearts (Fig. 2f). The mononucleated CMs ratio was further assessed by isolating CMs at P28, and the results showed that the percentage of mononucleated CMs was significantly decreased in Pex3-KO hearts compared to that in WT hearts (Fig. 2g). Subsequently, we assessed cardiac function at P1, P7, P14 and P28, which was significantly decreased in Pex3-KO mice decreased significantly at P28 (Fig. 2h, i). Survival curve analysis revealed that Pex3-KO mice did not show significant differences in mortality during development (Supplemental Fig. 3g). Taken together, CM-specific Pex3-KO disrupted endogenous proliferation and development of the neonatal myocardium, which affected cardiac morphology and function by reducing the number of CMs rather than altering cell size.

### Peroxisome-related metabolic dysfunction in the myocardium of Pex3-KO mice

To elucidate the specific mechanism underlying impaired endogenous myocardial proliferation resulting from PEX3 deletion, we extracted heart tissue from Pex3-KO and WT mice at P7 (the node of the neonatal myocardial regeneration timeline) for RNA-seq. The results showed 510 significantly upregulated genes and 116 downregulated considerably genes (Supplemental Fig. 4a). Subsequently, we performed gene function annotation (Gene Ontology) analysis on the differentially expressed genes between the two groups. The findings highlighted that differentially expressed genes were primarily enriched in peroxisome-related biological functions, notably in redox homeostasis, cellular metabolism, plasma membrane assembly and signaling[19] (Supplemental Fig. 4b). IF staining showed abnormal number and morphology of peroxisomes in Pex3-KO mice at P7 (Supplemental Fig. 4c). In addition, Transmission electron microscopy results also showed abnormal morphology and quantity of peroxisomes in Pex3-KO hearts at P28 (Supplemental Fig. 4d).

To further clarify the ROS metabolism disruption caused by PEX3 deficiency, we measured the signal intensity of Dihydroethidium (DHE) fluorescence under physiological conditions. IF results showed that the DHE fluorescent intensity was significantly increased in the Pex3-KO mice at P7 (Supplemental Fig. 4e). We also assessed the Malondialdehyde (MDA), a marker of lipid peroxidation used to evaluate ROS[20]. The results demonstrated a significant increase in the MDA level at P7 in Pex3-KO mice, indicating heightened ROS in the absence of PEX3 (Supplemental Fig. 4f). The expression level of γH2X (a marker of DNA

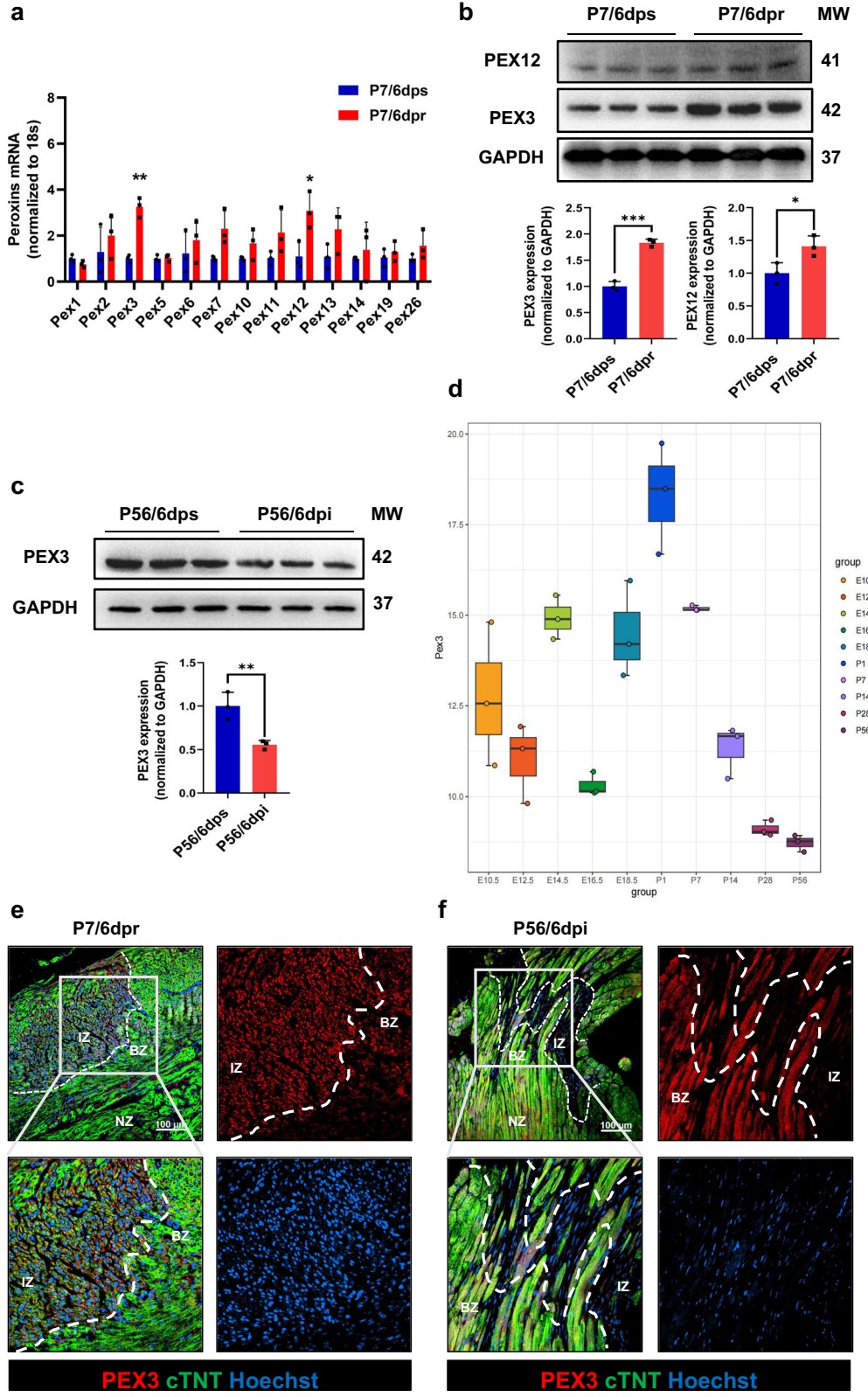

**Fig. 1 | PEX3-mediated peroxisome function is associated with regenerative repair after myocardial injury. a** mRNA level of peroxins genes in the mice heart. Expression of 13 classic peroxins was examined at P7/6dps and P7/6dpr. The fold change was calculated by normalizing to 18S ($n = 3$). **b** Western blotting and quantification analysis of PEX3 and PEX12 in P7/6 dps and P7/6 dpr hearts ($n = 3$). **c** Western blotting and quantification analysis of PEX3 at P56/6dps and P56/6dpi hearts ($n = 3$). **d** Relative expression levels of PEX3 in mice hearts at different developmental stages by public RNA-seq data (E10.5, E12.5, E14.5, E16.5, E18.5, P1, P7, P14, P28, P56). **e** Representative images of PEX3 (red) in P7/6 dpr myocardium. cTNT = Green, Hoechst = Blue. White dotted line separated the infarct zone from the infarct border zone and normal zone. Scale bar = 100 μm. **f** Representative images of PEX3 (red) in P56/6 dpi myocardium. cTNT = Green, Hoechst = Blue. White dotted line separated the infarct zone from the border zone and normal zone. Scale bar = 100 μm. Unpaired *t*-test applied for (**a–c**). Data shown as mean ± SEM. *$P < 0.05$, **$P < 0.01$, ***$P < 0.001$.

**Fig. 2 | Endogenous proliferation and development block in the myocardial physiological state of cardiomyocyte-specific PEX3 knockout mice.**
**a** Schematic illustration of the experimental design. **b**, **c** Immunofluorescence staining and quantification analysis of Ki67 and pH3 (green) in Pex3-KO mice from P1 to P28 were compared with WT mice (*n* = 6). cTNT = Red, Hoechst = Blue. Scale bar = 50 μm. (Ki67: 8293 CMs at P3 of WT mice group, 8802 CMs at P3 of Pex3-KO mice group; pH3: 14525 CMs at P3 of WT mice group, 14821 CMs at P3 of Pex3-KO mice group). **d** The proportions of mononucleated, bi-/multinucleated cardiomyocytes from P1 – P28 in WT and Pex3-KO mice (*n* = 6). **e** Heart weight/body weight ratio from P1 – P28 in WT and Pex3-KO mice (*n* = 6). **f** Representative heart images, Masson and wheat germ agglutinin (WGA, white) staining of myocardial slices from WT and Pex3-KO mice at P28. Hoechst=Blue. Scale bar = 1 mm, scale bar = 50 μm. **g** The proportions of mononucleated, bi-/multinucleated cardiomyocytes in WT and Pex3-KO mice were compared using isolated cardiomyocytes from P28 hearts and stained with cTNT (green) and DAPI (blue) (*n* = 6). Scale bar = 100 μm. **h** Echocardiography measurements of ejection fraction and fractional shortening in WT and Pex3-KO mice from P1 – P28 (*n* = 9). **i** Representative images of echocardiogram and comparison of ejection fraction and fractional shortening in WT and Pex3-KO mice at P28 (*n* = 9). Unpaired *t*-test applied for (**f**, **g**, and **i**). Two-way ANOVA and Tukey's Multiple Comparison Test were performed for **b**–**e**, **h**. Data shown as mean ± SEM. N.S, Not Significant, *P < 0.05, **P < 0.01, ***P < 0.001.

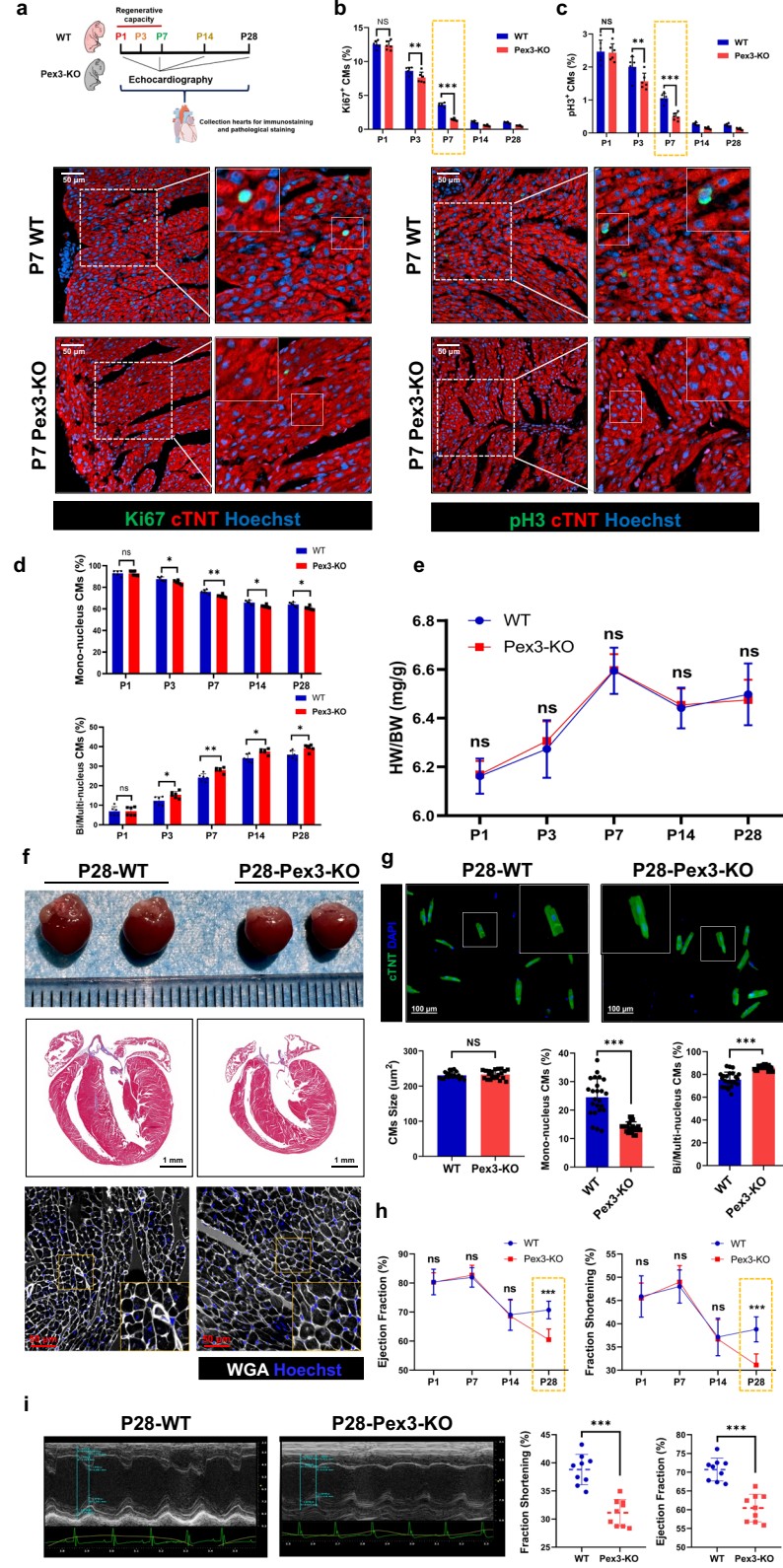

damage) in Pex3-KO mice was also significantly elevated at P7 (Supplemental Fig. 4g). These results provided further evidence that the deficiency of PEX3 disrupted peroxisome function, promoting myocardial ROS accumulation. These changes might contribute to the cessation of CMs cell cycle progression and the loss of cardiac regenerative capacity in the setting of PEX3 deficiency.

## PEX3 promotes CMs proliferation in vitro

The above results initially revealed the indispensable role of PEX3 in physiological myocardial proliferation and development. To eliminate potential interference from other cell types, we isolated neonatal mice CMs with a purity of about 90% in vitro to specifically explore the effect of PEX3 on CMs proliferation (Supplemental Fig. 5a). We employed PEX3 overexpression

and control adenovirus vectors with a CM-specific promoter (Cardiac Troponin T, Ad5: cTNT-). Similarly, we developed PEX3-knockdown and control CM-targeted vectors. Gradually varying concentrations of these vectors were utilized to determine the optimal transfection dose in CMs (MOI: 60) (Supplemental Fig. 5b). Firstly, WB analysis validated the intervention efficiency in CMs and Non-CMs (Fig. 3a). Furthermore, utilizing the Cell Counting Kit-8 (CCK8) assay, we demonstrated that the overexpression of PEX3 significantly enhanced the survival rate of CMs in vitro (Fig. 3b). Moreover, the total number of CMs increased considerably after the overexpression of PEX3 (Supplemental Fig. 5c, Fig. 3c). To further investigate the impact of PEX3 on CMs proliferation, we measured the levels of the cell proliferation indicators (Ki67, pH3, EdU and Aurora B). The IF results showed that the percentage of positively stained cells with Ad5: cTNT-PEX3 was significantly higher in the Ad5: cTNT-PEX3 group than in the Ad5: cTNT-CON group (Fig. 3d–g). To further investigate the impact of PEX3 on CMs under ischemia and hypoxia, we established the oxygen/glucose deprivation model and overexpressed PEX3 in vitro. We assessed the percentage of Tunel+ cells via IF, and the results showed that over-expression of PEX3 significantly inhibited the incidence of apoptosis (Fig. 3h). Subsequently, we quantified the levels of ROS using DCF fluorescence measurement, and IF analysis further corroborated that over-expression of PEX3 effectively reduced the ROS abundance in CMs (Fig. 3i). Flow cytometry assays revealed a notable accumulation of CMs in the S phase and G2 phase upon PEX3 overexpression (Fig. 3j).

Similarly, the intervention efficiency in CMs and Non-CMs was measured by WB analysis after transfecting CMs or Non-CMs with adenovirus vectors for CM-targeted PEX3 knockdown (Supplemental Fig. 5d). CCK-8 assays and the total number of CMs results showed that PEX3 knockdown decreased the number of viable CMs and the total CMs counts (Supplemental Fig. 5e, f). Moreover, our results demonstrated that in vitro, the percentage of CMs positive for cell proliferation indicators (Ki67, pH3, EdU and Aurora B) was significantly lower in the Ad5: cTNT-PEX3i group compared to the Ad5: cTNT-CONi group (Supplemental Fig. 5g–j). Conversely, the percentage of Tunel+ cells in the Ad5: cTNT-PEX3i group was significantly higher than that in the Ad5: cTNT-CONi group under oxygen/glucose deprivation conditions (Supplemental Fig. 5k). We also found that PEX3 knockdown increased ROS levels (Supplemental Fig. 5l). Finally, flow cytometry analysis revealed that PEX3 knockdown caused a decrease in the proportion of the S-phase and G2-phase CMs (Supplemental Fig. 5m). Collectively, the above results indicated that PEX3 promoted neonatal CMs proliferation in vitro.

### Deletion of PEX3 inhibits neonatal myocardial regeneration after AR

After investigating the beneficial effects of PEX3 on CM proliferation in vitro and observing that Pex3-KO under physiological conditions impeded myocardial endogenous proliferation and development, we were interested in further exploring the impact of PEX3 during myocardial injury in vivo. We performed AR at P1 of Pex3-KO and WT mice (Fig. 4a). The expression of PEX3 was determined by WB analysis at 6 days post-AR (dpr) (Fig. 4b). Furthermore, we conducted IF staining (Ki67, pH3, EdU and AuroraB) on tissue samples from the border zone surrounding the apical injury in both groups of mice. The Pex3-KO group exhibited significantly lower positivity for myocardial proliferation signals in the injury area than the WT group (Fig. 4c–f). Furthermore, we observed a decrease in the proportion of mononucleated CMs, accompanied by a relative increase in binucleated CMs following PEX3 knockdown (Fig. 4g). Additionally, we measured the levels of DHE and MDA at 6dpr, and the results indicated that PEX3 knockdown led to increased oxidative stress (Fig. 4h, i). Detection of γH2X also showed that DNA damage was aggravated under PEX3 knockdown conditions (Supplemental Fig. 6a, Fig. 4j). Notably, no significant difference was observed between the Pex3-KO group and the WT group regarding CMs size and heart weight/body weight at 28dpr (Fig. 4k, l). To further clarify the functional impact of PEX3 knockdown after AR, we performed echocardiography at 1dpr and 28dpr, which showed that PEX3

knockdown led to a decrease in cardiac function at 28dpr (Fig. 4m). Moreover, Masson staining showed that PEX3 knockdown inhibited myocardial regeneration (Fig. 4n). Survival curve analysis revealed that PEX3 knockdown did not increase mortality during AR between the Pex3-KO and WT groups (Fig. 4o). Thus, the results in vivo demonstrated that PEX3 was indispensable for myocardial regeneration after AR in neonatal mice.

### PEX3 promotes CMs proliferation and improves cardiac function in adult mice following MI

Because the fragile proliferative capacity in the adult myocardium does not allow significant replenishment of the injured myocardium, the adult myocardium fills with scar tissue after MI, and cardiac function declines[21]. Consequently, to investigate whether PEX3 impacted CMs survival and regenerative repair after MI in adult mice, we employed homogeneous positional ligation of the left anterior descending coronary artery to induce MI in P56 mice. Furthermore, multipoint in situ injection of AAV9: cTNT-PEX3 or AAV9: cTNT-CON was performed, and the effectiveness of transfection was confirmed through IF and WB analysis at 14 days post infarction (dpi) (Supplemental Fig. 7a, Fig. 5a, b). IF staining of proliferation indicators, including Ki67, pH3, EdU and Aurora B, was performed on both groups of mice. Remarkably, the results demonstrated that PEX3 over-expression significantly enhanced myocardial proliferation in the border zone surrounding the injury in adult mice compared to that in the control group (Fig. 5c–f). In addition, we performed Tunel staining in the infarct border zone at 14 dpi, and the results showed that the apoptosis level of the PEX3 group was significantly lower than that of the CON group (Fig. 5g). Furthermore, we observed an increase in the proportion of mononucleated CMs and a decrease in the proportions of binucleated CMs following PEX3 overexpression (Fig. 5h). Additionally, our DHE, MDA and γH2X measurement results indicated that PEX3 overexpression reduced oxidative stress levels at 14 dpi, consequently protecting myocardial tissues from DNA damage (Supplemental Fig. 7b, Fig. 5i–k). The CMs size, as assessed by WGA staining, did not show significant differences between the PEX3 and the CON groups at 28 dpi (Fig. 5l). Subsequently, echocardiography results revealed that PEX3 overexpression improved cardiac function at 28dpi (Fig. 5m). Additionally, Masson staining demonstrated that PEX3 over-expression significantly reduced the infarct size (Fig. 5n). Finally, survival rate analysis revealed that PEX3 overexpression played a protective role during the MI process in both groups, although no significant difference was observed (Fig. 5o).

To further investigate whether systemic injection of AAV9-cTnT-PEX3 could promote the CMs proliferation after MI, we constructed MI models in adult mice and AAV9-cTnT-PEX3 (AAV9-cTnT-CON) was injected by tail vein at 3 dpi. IF staining of proliferation indicators (Ki67, pH3 and Aurora B) was performed on two groups of mice at 14 dpi. The results demonstrated that PEX3 overexpression by tail vein injection significantly enhanced CMs proliferation in the border zone surrounding the injury and normal zone of adult mice compared to that in the control group (Supplemental Fig. 7c–e).

Similarly, we constructed MI models in WT and Pex3-KO adult mice. At 14dpi, we detected proliferation indicators (Ki67 and pH3) and apoptosis indexes (Tunel+), which revealed a lower CMs proliferation rate in the Pex3-KO group than in the WT group. In contrast, the level of apoptosis was significantly higher in the Pex3-KO group (Supplemental Fig. 8a–c). We measured CMs size via WGA staining and found no difference between the two groups, while a significant decrease in the proportion of mononucleated CMs and an increase in binucleated CMs were observed in the Pex3-KO group compared to the WT group (Supplemental Fig. 8d, e). Echocardiography showed that the cardiac function of the Pex3-KO group was significantly worse than that of the WT group (Supplemental Fig. 8f). Whereas Masson staining indicated that PEX3 knockdown significantly increased the infarct size at 28 dpi (Supplemental Fig. 8g). In addition, the survival rate of mice in the Pex3-KO group after MI was significantly lower than that in the WT group (Supplemental Fig. 8h).

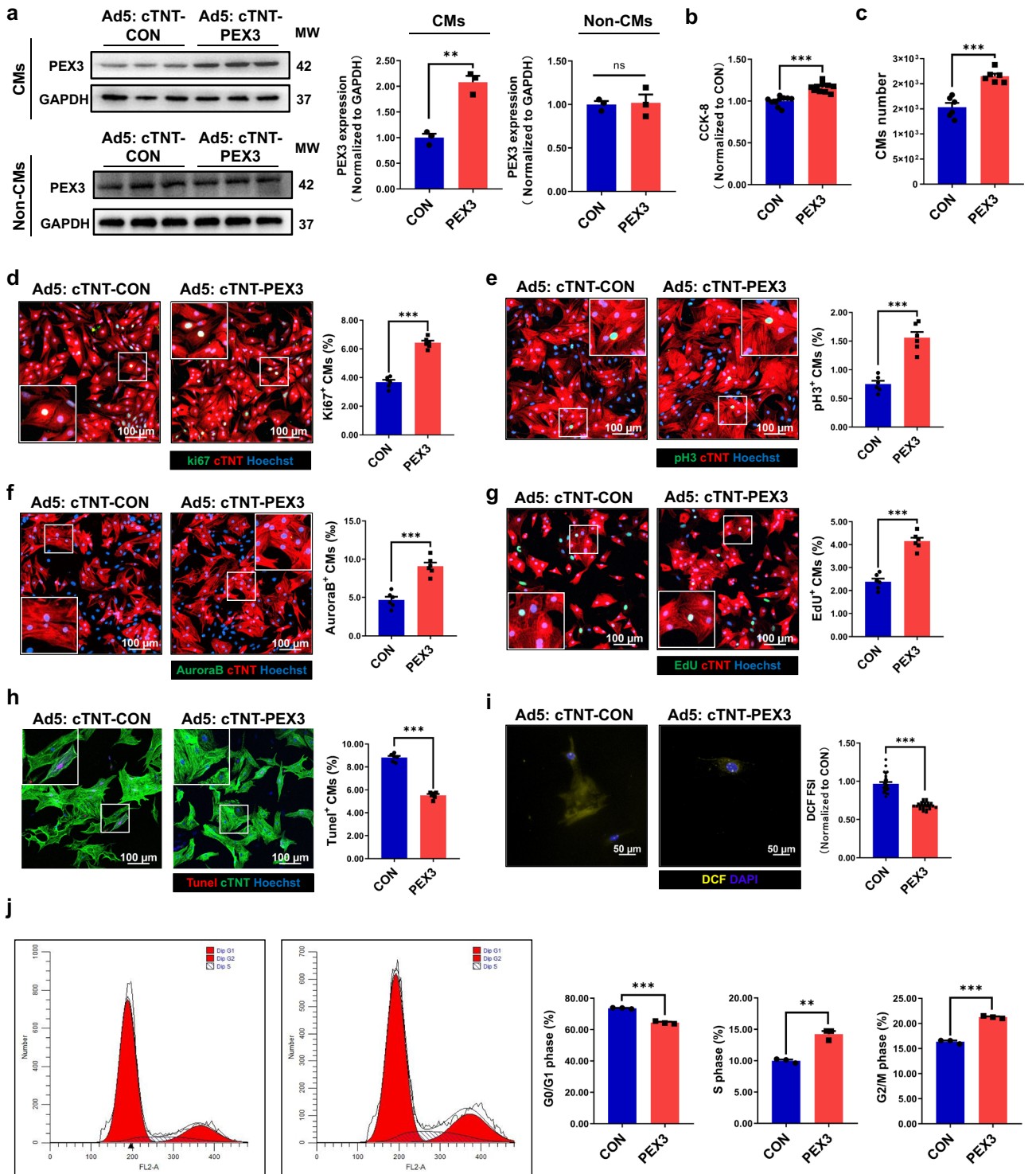

**Fig. 3 | PEX3 promotes cardiomyocytes proliferation in vitro. a** Western blotting and quantification analyses of PEX3 protein levels in primary cardiomyocytes and Non-CMs transfected with Ad5-cTNT-PEX3 and Ad5-cTNT-CON (*n* = 3). **b** CMs viability quantified between Ad5-cTNT-PEX3 groups and Ad5-cTNT-CON groups by CCK-8 assay (*n* = 10). **c** The total number of CMs between Ad5-cTNT-PEX3 groups and Ad5-cTNT-CON groups (*n* = 6). **d–g** IF staining and quantification analysis of Ki67, pH3, Aurora B and EdU (green) after transfection with Ad5-cTNT-PEX3 or Ad5-cTNT-CON in CMs (*n* = 6). cTNT = Red, Hoechst=Blue. Scale bar = 100 μm. (Ki67: 2478 CMs in the Ad5-cTNT-CON group, 2534 CMs in the Ad5-cTNT-PEX3 group; pH3: 2639 CMs in the Ad5-cTNT-CON group, 2263 CMs in the Ad5-cTNT-PEX3 group; Aurora B: 2970 CMs in the Ad5-cTNT-CON group,

2897 CMs in the Ad5-cTNT-PEX3 group; EdU: 2738 CMs in the Ad5-cTNT-CON group, 2486 CMs in the Ad5-cTNT-PEX3 group). **h** IF staining of Tunel (red) in the PEX3 groups and CON groups post oxygen/glucose deprivation treatment (*n* = 6). cTNT = Green, Hoechst = Blue. Scale bar = 100 μm. (Tunel: 3637 CMs in the Ad5-cTNT-CON group, 3610 CMs in the Ad5-cTNT-PEX3 group). **i** IF staining and quantification analysis of DHE (yellow) in PEX3 groups compared with CON groups (*n* = 6). DAPI=Blue. Scale bar = 50 μm. **j** Flow cytometry analysis of primary CMs transfected with Ad5-cTNT-PEX3 or Ad5-cTNT-CON (*n* = 3). Unpaired *t*-test applied for (**a–j**). Data shown as mean ± SEM. N.S, Not Significant, **P < 0.01, ***P < 0.001.

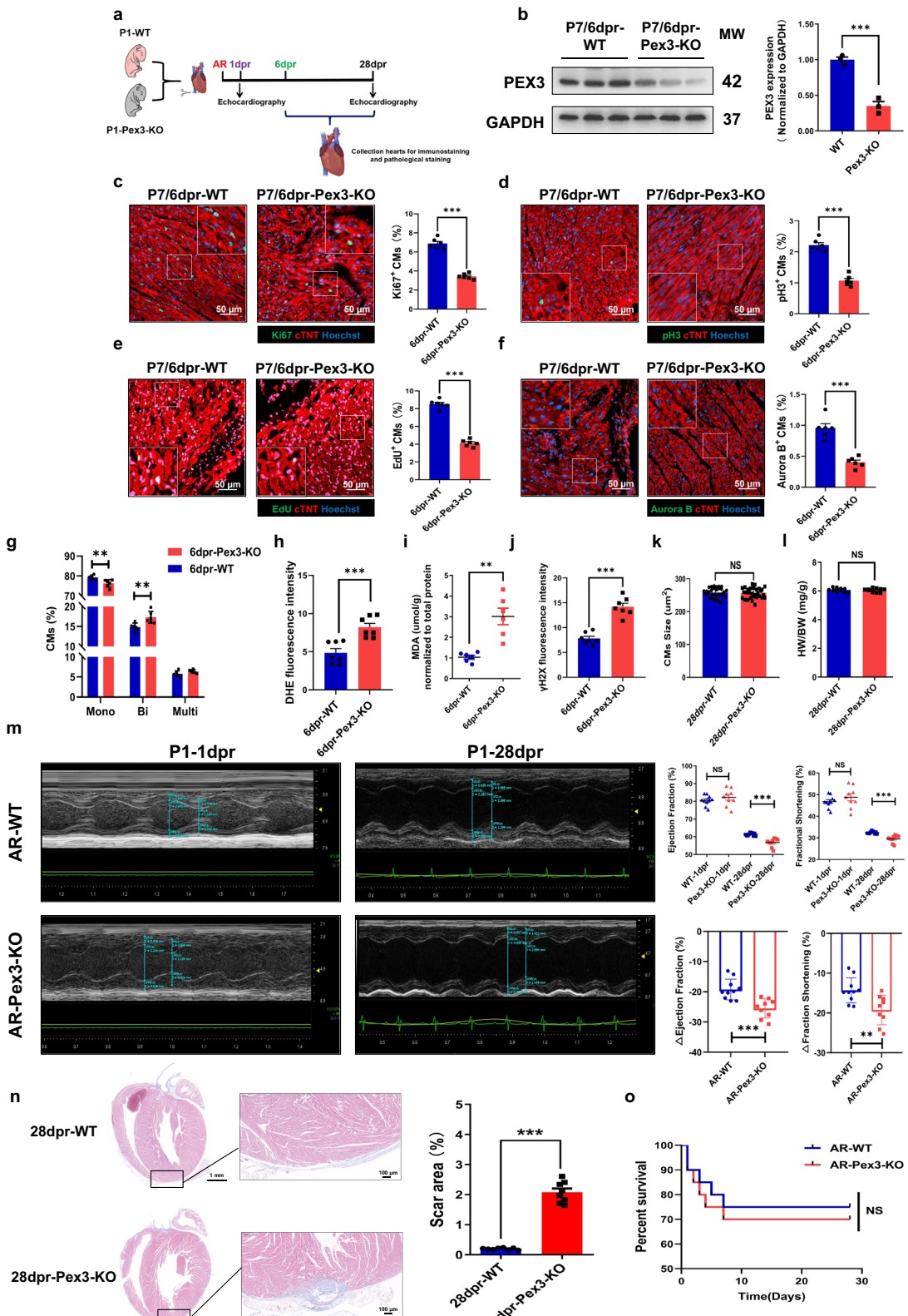

## Supplement with plasmalogen ameliorates impaired myocardial regenerative repair in Pex3-KO mice

Previously, we investigated the impact of PEX3-mediated peroxisome function on oxidative stress metabolism following MI, and we found that it plays a crucial role in promoting myocardial regenerative repair. The peroxisome participates in various aspects of disease progression, including regulating intracellular lipid metabolism, fatty acid oxidation and lipid synthesis. Furthermore, it plays a significant role in producing plasmalogen, which is essential to cellular functions and signaling pathways[22,23]. To investigate whether lipid metabolism mediated the effect of PEX3 on regenerative repair after myocardial injury, we performed lipidomics analysis on Pex3-KO and WT mice at 14 dpi. The obtained lipidomic data were

**Fig. 4 | Deletion of PEX3 inhibits neonatal myocardial regeneration after apical resection. a** Schematic illustration of the experimental design. **b** Western blotting and quantitative analysis of PEX3 protein levels in neonatal mice at 6 dpr ($n = 3$). **c–f** Effect of PEX3 on cell cycle markers (Ki67, pH3, EdU and Aurora B) (green) in neonatal mice at 6 dpr in the WT and Pex3-KO mice ($n = 6$). cTNT = Red, Hoechst = Blue. Scale bar = 50 μm. (Ki67: 5735 CMs in the WT group, 8093 CMs in the Pex3-KO group; pH3: 10659 CMs in the WT group, 12818 CMs in the Pex3-KO group; Aurora B: 6973 CMs in the WT group, 6468 CMs in the Pex3-KO group; EdU: 5116 CMs in the WT group, 7891 CMs in the Pex3-KO group). **g** The proportions of mononucleated, binucleated and multinucleated cardiomyocytes between Pex3-KO groups and WT groups at 6 dpr ($n= 6$). **h** Quantification analysis of DHE in Pex3-KO groups compared with WT groups at 6 dpr ($n = 7$). **i** The level of MDA between Pex3-KO groups and WT groups at 6dpr ($n = 6$). **j** Quantification analysis of γH2X between Pex3-KO groups and WT groups at 6 dpr ($n = 7$). **k** Quantification analysis of WGA staining in Pex3-KO groups compared with WT groups at 28dpr ($n = 6$). **l** Heart weight/body weight ratio between Pex3-KO groups and WT groups at 28dpr ($n = 10$). **m** Echocardiography measurements of ejection fraction and fractional shortening between Pex3-KO groups and WT groups at 1dpr and 28dpr ($n = 9$-$10$). **n** Masson staining and quantification of scar tissue in the resected apex of heart between Pex3-KO groups and WT groups ($n = 8$). Scale bar = 100 μm, scale bar = 1 mm. **o** Survival rate of mice in Pex3-KO groups and WT groups from 1dpr to 28 dpr ($n = 20$). Unpaired $t$-test applied for (**b–n**). Kaplan–Meier (log-rank test) was performed for o. Data shown as mean ± SEM. N.S, Not Significant, **$P < 0.01$, ***$P < 0.001$.

then subjected to clustering and labeling based on the Class and SubClass categories of the LIPID MAPS® system nomenclature. By employing this analytical strategy, we aimed to elucidate the specific changes in lipid metabolism associated with PEX3 deficiency and their potential implications in myocardial injury and repair processes. We further selected statistically different ($P < 0.05$) lipid metabolites for clustering and labeling. The results showed that part of plasmalogens (alkenyl-acyl-PE/PC) and degradation products (monoalkenyl-LPE/LPC, monoacyl-LPI/LPE/LPC) were significantly reduced in Pex3-KO mice after MI (Fig. 6a, b).

Plasmalogen is a class of membrane glycerophospholipids with unique properties characterized by a vinyl ether bond at the sn-1 position of the glycerol backbone. The characteristics of the vinyl ether bond enable plasmalogen to possess potential antioxidant functions[24,25]. Cardiac tissue is abundant in plasmalogen, particularly those with a choline or ethanolamine headgroup[26]. Plasmalogen in the heart has been found to play a specific role in regulating the levels of cardiolipin, as a heart-specific phospholipid[27]. We supplemented plasmalogen using Alkylglycerol (AKG) and analyzed the corresponding changes in plasmalogen through lipidomics analysis[28] (Fig. 6c, d). Subsequently, we detected proliferation indicators (Ki67 and pH3) in the border zone at 14dpi, and the results showed that endogenous supplementation with plasmalogen rescued the CMs proliferation impaired by Pex3-KO (Fig. 6e, f). We employed echocardiography to assess the ejection fraction and fractional shortening and found that no significant difference in the cardiac function among all groups at 1dpi, while supplementation with plasmalogen improved cardiac function in Pex3-KO mice at 28 dpi (Supplemental Fig. 9a, Fig. 6g). In addition, Masson staining showed that plasmalogen supplementation could reduce the infarct size at 28dpi (Fig. 6h). These findings suggested that supplementation with plasmalogen could improve cardiac function and reduce the infarct size, thereby promoting myocardial repair in the context of PEX3 deficiency.

## PEX3 interferes with ITGB3 membrane localization through plasmalogen metabolism

To define the potential working mechanism of PEX3 on promoting myocardial repair, we isolated the myocardial plasma membrane proteins of adult mice following myocardial infarction in Pex3-KO mice and WT mice at P56, and identified them by subcellular markers (cell membrane marker-ATP1A3, cytoplasmic maker-GAPDH, and nuclear marker-HH3) to detect differentially expressed proteins at the cell membrane by proteomics (Supplemental Fig. 10a, Fig. 7a). We selected proteins that were relatively downregulated and significantly differentially expressed in the Pex3-KO myocardial plasma membrane ($p < 0.05$, fold change (FC) > 1.2) for Gene ontology analysis and found that the differentially expressed proteins were enriched in integrin-associated alterations of the membrane structure, in addition to myogenic proliferation and wound healing (Supplemental Fig. 10b, Fig. 7b). We further analyzed the network topology of critical proteins among the differentially expressed plasma membrane proteins using the Hubba plugin for Protein-Protein Interaction networks analysis. Integrin beta 3 (ITGB3) was at the core and scored the highest among the top 15 (Fig. 7c, Supplemental Table 2). ITGB3, one of the most widely studied members in the integrin family, induces myocardial regeneration in

CMs near the injury site in zebrafish[29]. Previous studies also highlighted the role of ITGB3 in cardiosphere-mediated myocardial regeneration, a process involved in the regeneration and repair of cardiac tissue[30]. We found that ITGB3 expression was decreased in Pex3-KO mice myocardial tissue through IF staining (Fig. 7d). Subsequently, our results demonstrated that overexpression of PEX3 led to an increase in the plasma membrane localization of ITGB3, while overexpression or knockdown of ITGB3 did not affect the expression of PEX3 and PMP70 (Supplemental Fig. 10c–f). Next, we performed a combination experiment with AAV9: cTNT-PEX3 or/and ITGB3i after MI, and the WB assay results showed that ITGB3 inhibition significantly attenuated the increase in ITGB3 expression at the cell membrane caused by PEX3 (Supplemental Fig. 11a, b). Furthermore, IF staining (Ki67 and pH3) indicated that the inhibition of ITGB3 attenuated the beneficial effects of PEX3 in promoting myocardial regeneration following MI (Supplemental Fig. 11c, d).

Plasmalogen, a subtype of ether lipid, has been indicated to participate in membrane biology by regulating lipid raft microdomains. It is possible that the function and properties conferred by PEX3 influenced the process of myocardial regenerative repair through their involvement in membrane biology and cell signaling. To investigate whether the regulatory effect of PEX3 on plasma membrane-localized ITGB3 was mediated through plasmalogen metabolism, we conducted a combination experiment on Pex3-KO and WT mice fed with AKG diets after MI. Our findings revealed that supplementation with plasmalogen improved the localization of ITGB3 in the plasma membrane of Pex3-KO mice at 14 dpi (Fig. 7e). Meanwhile, we observed changes in the plasma membrane by transmission electron microscopy and found that supplementation of Pex3-KO mice with plasmalogen attenuated the disturbance of the plasma membrane structure at 14dpi (Fig. 7f). Finally, we combined AGK and AAV9: cTNT-ITGB3i in the Pex3-KO mice, and verified that ITGB3 knockdown effectively inhibited the localized expression of ITGB3 in the plasma membrane after plasmalogen supplementation (Fig. 7g, h). We further assessed the effects of ITGB3 knockdown on myocardial regenerative capacity by examining the proliferation indicators Ki67 and pH3. Our findings demonstrated that ITGB3 knockdown reversed the restoration of CMs proliferation in Pex3-KO mice induced by plasmalogen supplementation at 14dpi (Fig. 7i, j). Echocardiography results showed that the cardiac function has no significant difference among all groups at 1dpi, while ITGB3 knockdown could reverse the improvement in cardiac function and reduction in infarct size in Pex3-KO mice with plasmalogen supplementation at 28 dpi (Supplemental Fig. 11e, Fig. 7k, l). These findings suggested that ITGB3 played a critical role in mediating the regenerative effects of plasmalogen supplementation in the context of PEX3 deficiency.

## PEX3 regulates the ITGB3-mediated AKT/GSK3β pathway to promote myocardial regenerative repair

Previous research has established a strong association between ITGB3 and the activation of the AKT signaling pathway[31,32]. Moreover, the Kyoto Encyclopedia of Genes and Genomes analysis revealed significant enrichment of the AKT signaling pathway for the differentially expressed plasma membrane proteins (Fig. 8a). Therefore, we first verified the expression of

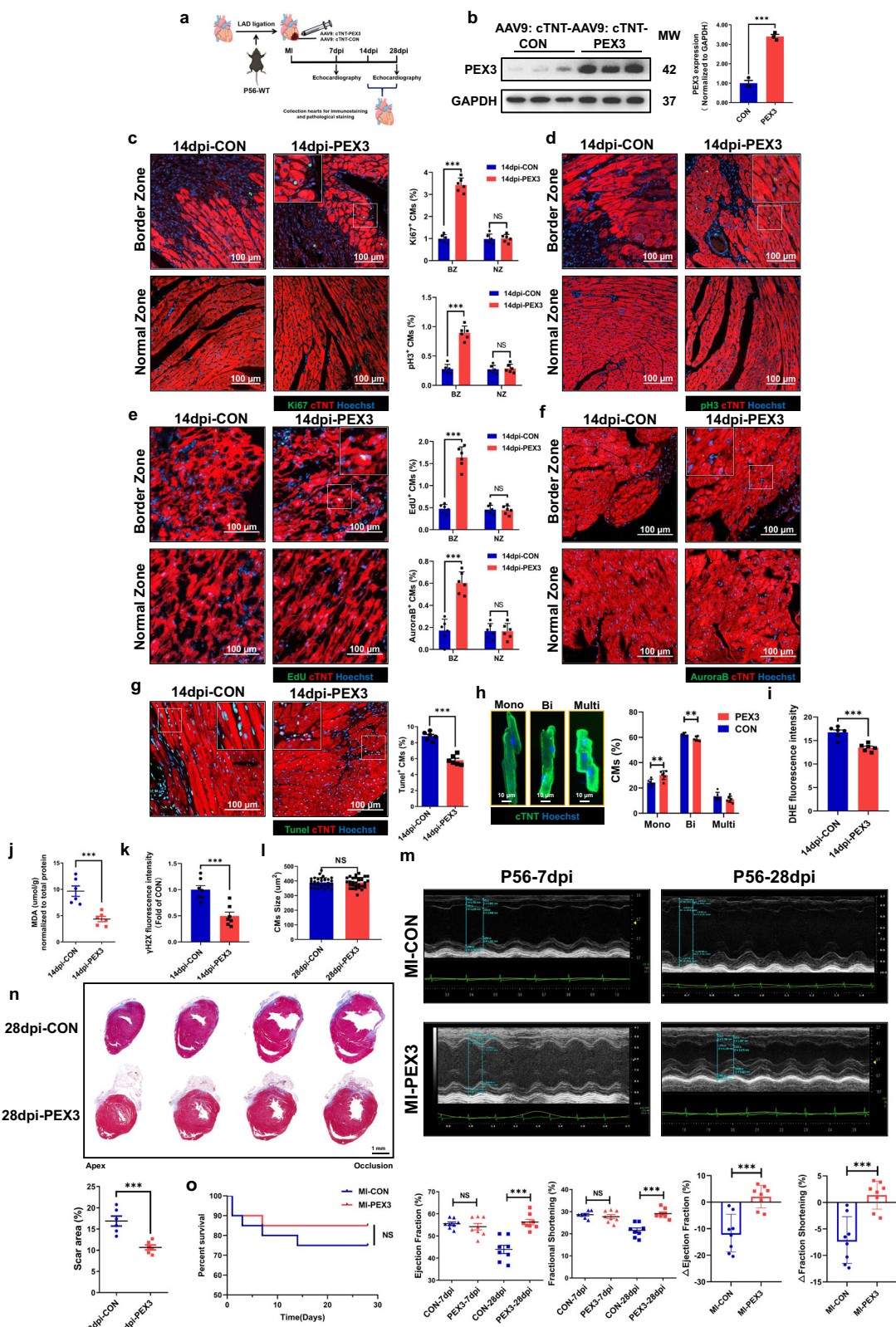

AKT signaling pathway members in Pex3-KO and WT mice. The WB results showed decreased phosphorylation at Thr308 and Ser473 of AKT as well as inhibited activation of downstream p-GSK3β in the Pex3-KO myocardium, whereas there were no significant alterations in the upstream targets of AKT (Fig. 8b). Numerous studies have provided evidence supporting the role of the AKT signaling pathway in mediating myocardial

regenerative repair. Specifically, ITGB3 has been identified as a direct participant in the activation of AKT within the plasma membrane, functioning as an effector receptor protein[33–35]. To further investigate the signaling pathway obtained from Kyoto Encyclopedia of Genes and Genomes analysis and to determine any potential alterations in the subcellular localization of the phosphorylation sites (Thr308 and Ser473) of AKT in Pex3-KO mice

**Fig. 5 | PEX3 promotes CMs proliferation and improves cardiac function in adult mice following MI. a** Schematic illustration of the experimental design. **b** Western blotting and quantitative analysis of PEX3 protein levels in adult mice at 14 dpi after MI and AAV9-cTNT-PEX3 or AAV9-cTNT-CON injection ($n = 3$). **c–f** Effect of PEX3 on cell cycle markers (Ki67, pH3, EdU and Aurora B) (green) in adult mice after MI and AAV9-cTNT-PEX3 or AAV9-cTNT-CON injection at 14 dpi ($n = 6$). cTNT = Red, Hoechst = Blue. Scale bar = 100 μm. (Border Zone: Ki67: 4796 CMs in the AAV9-cTNT-CON group, 3678 CMs in the AAV9-cTNT-PEX3 group; pH3: 4275 CMs in the AAV9-cTNT-CON group, 4977 CMs in the AAV9-cTNT-PEX3 group; Aurora B: 2511 CMs in the AAV9-cTNT-CON group, 2827 CMs in the AAV9-cTNT-PEX3 group; EdU: 3220 CMs in the AAV9-cTNT-CON group, 3855 CMs in the AAV9-cTNT-PEX3 group). **g** IF staining and quantification of Tunel signals (green) in the PEX3 groups and CON groups at 14 dpi after MI ($n = 6$). cTNT = Red, Hoechst = Blue. Scale bar = 100 μm. (Tunel: 4940 CMs in the AAV9-

cTNT-CON group, 4024 CMs in the AAV9-cTNT-PEX3 group). **h** Isolation and quantification of the mononucleated, binucleated and multinucleated cardiomyocytes number between PEX3 groups and CON groups at 14 dpi ($n = 6$). cTNT = Green, Hoechst = Blue. Scale bar = 10 μm. **i** Quantification analysis of DHE in PEX3 groups compared with CON groups at 14dpi ($n = 6$). **j** The level of MDA between PEX3 groups and CON groups at 14dpi ($n = 6$). **k** Quantification analysis of γH2X in PEX3 groups and CON groups at 14 dpi ($n = 7$). **l** Quantification analysis of WGA in PEX3 groups and CON groups at 28 dpi ($n = 6$). **m** Echocardiography measurements of ejection fraction and fractional shortening between PEX3 groups and CON groups at 7dpi and 28dpi ($n = 8$). **n** Masson staining and quantification of scar tissue in PEX3 groups and CON groups at 28 dpi ($n = 6$). Scale bar = 1 mm. **o** Survival rate of mice in PEX3 groups and CON groups from 1dpi to 28 dpi ($n = 20$). Unpaired $t$-test applied for (**b–n**). Kaplan–Meier (log-rank test) was performed for (**o**). Data shown as mean ± SEM. N.S, Not Significant, $**P < 0.01$, $***P < 0.001$.

after MI, we conducted cytoplasm membrane separation with myocardial tissues. The results revealed a distinct impairment in the phosphorylation of AKT at the membrane of Pex3-KO mice at 14dpi (Fig. 8c, Supplemental Fig. 12a). Likewise, our in vitro experiments demonstrated notable activation of the AKT/GSK3β signaling pathway and its downstream effectors upon overexpression of PEX3 (Supplemental Fig. 12b, c, Fig. 8d). To investigate whether ITGB3/AKT/GSK3β serves as a downstream effector of PEX3 in mediating myocardial regenerative functions, we conducted combination experiments in vitro. These experiments involved targeted knockdown of ITGB3 in CMs, followed by the administration of SC79 (an AKT agonist) or TDZD8 (a GSK3β inhibitor) after the overexpression of PEX3 (Supplemental Fig. 12d, e). Subsequently, the assessment of proliferation indicators (Ki67 and pH3) showed that the adverse effects of ITGB3 knockdown on CMs proliferation could be reversed through interference with AKT or GSK3β (Supplemental Fig. 12f, g).

Finally, we aimed to explore whether PEX3 regulates the ITGB3-mediated AKT/GSK3β signaling pathway by influencing the metabolism of plasmalogen. We supplemented plasmalogen via AKG while simultaneously intervening with ITGB3 and the AKT/GSK3β pathway in Pex3-KO mice (Supplemental Fig. 12h, i, Fig. 8e). IF detection of proliferative indicators (Ki67 and pH3) revealed a sequential effect on CMs proliferation among plasmalogen, ITGB3, and the AKT/GSK3β pathway in Pex3-KO mice at 14dpi (Fig. 8f, g). Additionally, echocardiography and Masson staining further confirmed that the AKT/GSK3β pathway intervention reversed the ITGB3 knockdown-mediated functional and structural deterioration in Pex3-KO mice supplemented with plasmalogen (Fig. 8h, i). Above, these results showed that PEX3 promoted myocardial regeneration and improved cardiac function through plasmalogen metabolism-mediated ITGB3/AKT/GSK3β signaling activation (Fig. 8j).

## Discussion

Growing evidence demonstrates that the peroxisome, formerly regarded as the 'Ugly Duckling' among known organelles, has evolved into a multifunctional global participant with profound relevance for the health and disease of animal and plant organisms[23]. Additionally, studies have revealed that oxidative stress-induced DNA damage resulting from ROS accumulation plays a crucial role in the cessation of neonatal myocardial proliferation and that reducing ROS levels can partially restore myocardial regenerative capacity[7,36]. Therefore, peroxisome seems to have intricate connections with tissue regeneration and repair after injury. PEX3, a vital protein responsible for peroxisomal membrane synthesis, plays an extensive role in peroxisome generation, maturation, and clearance after aging[37]. M. S. Dahabieh et al. found that PEX3, PEX16, and PEX19 promoted tumorigenesis. Therefore, based on the peroxisome-related metabolic functions, PEX3 might be a valuable indicator for investigating ROS in myocardial regeneration. We initially found that the expressions of PEX3 and PEX12 were both increased after neonatal myocardial injury, and the increase of PEX3 was more significant. It is known that the most severe peroxisome-deficient phenotype was in yeast PEX3 cells, in which most of the

peroxisome matrix and membrane proteins were mislocalized[38]. Whereas PEX12 is a part of genes that encode peroxins involved in matrix protein import. Mutation in PEX12 results in smaller, spherical peroxisomal structures[39]. In polymorphic H.PEX3 cells, only a fraction of the peroxisome membrane proteins is present on peroxisomal membrane structures. These proteins are stable relative to other peroxisome membrane proteins, which are also incorrectly localized to the cytoplasm[40]. As a result, the different peroxins acting on matrix protein import and peroxisomal membrane biogenesis are often unable to be aligned, and this may behave differently in different model organisms[41]. In general, PEX3 is a prerequisite for peroxisomal regulation and served as a critical signaling in this study. Next, we further explored whether PEX3 is valuable in myocardial regeneration.

In redox homeostasis and lipid metabolism involving ether phospholipids (e.g., plasmalogen), bile acids and branched-chain fatty acids, peroxisome acts not only as passive executors but also as signal hubs, integrating information from multiple metabolic reactions[42]. Specifically, alterations in peroxisomal lipid generation disrupt the balance between structural and signaling effects, thereby promoting tissue proliferation[43,44]. In our research, a distinct deficiency in the synthesis of plasmalogen was observed, which is peroxisome-derived ether glycerophospholipids, in Pex3-KO mice, as evidenced by lipidomics analysis. Previous comprehensive lipidomics investigations involving multiple human populations revealed a significant negative correlation between plasmalogen and cardiovascular diseases[45], and the alteration of plasmalogen represented distinctive change in lipid metabolism during tissue regeneration[46,47]. Several studies have indicated the therapeutic potential of plasmalogen supplementation in degenerative and metabolic diseases through dietary interventions, providing evidence supporting the role of plasmalogen in metabolic regulation and the promotion of organismal homeostasis. However, these studies underperformed according to accurately measuring plasmalogen in targeted tissues after supplementation, which limited the interpretability of their findings[48–50]. We conducted plasmalogen supplementation using various precursor forms through AKG and assessed the effects using lipidomics analysis. Similar to findings from other studies, our results revealed that the plasmalogen in myocardial tissue predominantly consists of phosphatidylethanolamine and phosphatidylcholine species, with minor amounts of CL and PG detected[51]. Furthermore, compared to WT mice, there was a significant decrease in certain plasmalogen species in Pex3-KO mice, particularly those with sn-1 alkyl chains predominantly composed of 16:0, 18:0 or 18:1, and sn-2 acyl chains consisting mainly of polyunsaturated fatty acid 22:6. In our study, AKG therapy achieved widespread but heterogeneous supplementation of plasmalogen in myocardial tissue, including the species mentioned above. Similarly, Paul et al. found that AKG treatment exerted more pronounced effects on plasmalogen in plasma and adipose tissue, while the enhancement of plasmalogen in the liver and skeletal muscle was less noticeable[28]. The plasmalogen level appears to be more strictly regulated in the substantial organs. Therefore, further investigation into the effects of specific modulation of certain plasmalogen species in the heart may require adjustments in the ratio and

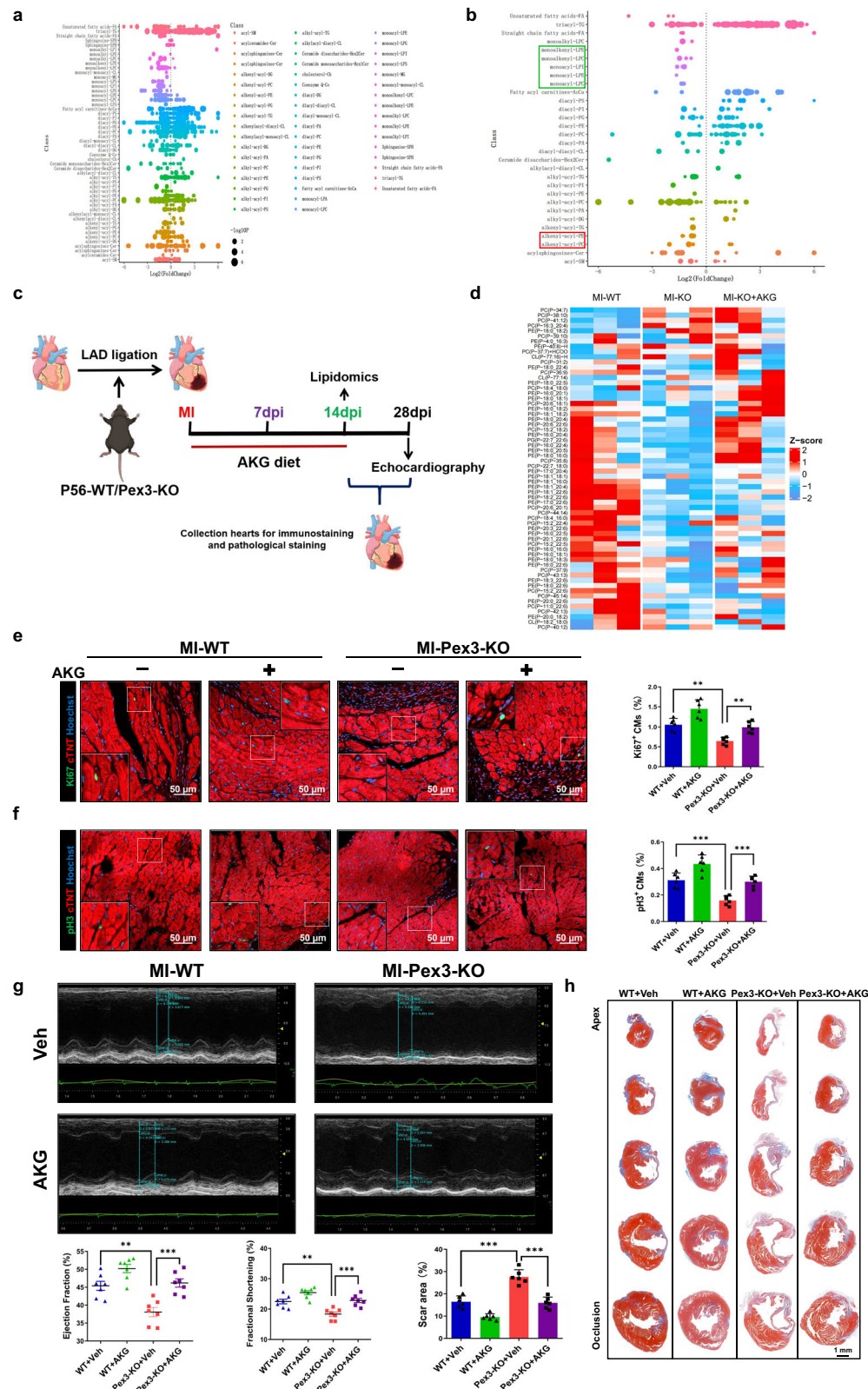

duration of AKG treatment. In conclusion, further research is needed to explore whether the heterogeneity of different head groups and fatty acid compositions at sn-1/sn-2 positions of plasmalogen plays different biological roles in the myocardium.

As significant constituents of cellular membranes, plasmalogen plays a crucial role in membrane biology, and changes in their proportion and content can influence the physical properties of cell membranes and dynamic processes such as membrane transport[52,53]. Therefore, they have the potential to impact various aspects of heart development, structure, and function. For instance, a deficiency of ether phospholipids in cardiac tissue can lead to impaired cardiac electrical coupling due to reduced levels of membrane proteins such as connexin 43[54]. Abnormalities in cell membrane

**Fig. 6 | Supplement of plasmalogen ameliorates impaired myocardial regenerative repair in Pex3-KO mice. a** Scatter diagram of the lipid metabolites in WT and Pex3-KO myocardium after MI by lipidomics analysis. **b** Scatter diagram of the differential lipid metabolites (filtered with $P < 0.05$) by lipidomics analysis (plasmalogen metabolites-green box, plasmalogen-red box). **c** Schematic illustration of the experimental design. **d** Clustered heatmap of the differentially expressed plasmalogens in WT+MI, Pex3-KO+MI and Pex3-KO+MI+AKG by lipidomics analysis (14dpi). **e, f** Representative images and quantification analysis of Ki67 and pH3 (green) in WT and Pex3-KO mice after MI and treated with AKG for 2 weeks ($n = 6$). cTNT = Red, Hoechst = Blue. Scale bar = 50 μm. (Ki67: 3538 CMs in the WT

+Veh group, 3585 CMs in the WT+AKG group, 3500 CMs in the Pex3-KO+Veh group, 4124 CMs in the Pex3-KO+AKG group; pH3: 4260 CMs in the WT+Veh group, 4722 CMs in the WT+AKG group, 4645 CMs in the Pex3-KO+Veh group, 5428 CMs in the Pex3-KO+AKG group). **g** Echocardiography measurements of ejection fraction and fractional shortening in WT and Pex3-KO mice after MI and treated with AKG for 2 weeks at 28 dpi ($n = 7$). **h** Masson staining and quantification of scar tissue in the injured heart of WT and Pex3-KO mice after MI and treated with AKG for 2 weeks at 28dpi ($n = 6$). Scale bar=1mm. Two-way ANOVA and Tukey's Multiple Comparison Test were performed for (**e–h**). Data shown as mean ± SEM. $**P < 0.01$, $***P < 0.001$.

signaling associated with ether phospholipids have been implicated in the occurrence and progression of various heart diseases[55,56]. Consequently, we opted to investigate the plasma membrane alteration of the receptor protein ITGB3 as the downstream target to further explore the regulatory role of PEX3 in plasmalogen metabolism, which aligned with the findings from RNA-seq analysis in Pex3-KO mice. It also seems to be a possible explanation for how ether phospholipids safeguard cells from apoptosis by modulating the AKT signaling pathway[57]. ITGB3 is a member of the integrin protein family, acting as a transmembrane receptor that mediates cell-cell and cell-extracellular matrix interactions. It promotes signal transduction through pathways such as AKT to stimulate proliferation, survival, and migration in various cell types[35,58,59]. In our research, we observed that specifically silencing ITGB3 in the myocardium disrupted the beneficial effects of PEX3-regulated plasmalogen metabolism on cardiac regenerative repair. Furthermore, by blocking multiple targets within the AKT/GSK3β signaling pathway, we further confirmed the crucial role of ITGB3 in activating AKT signaling at the plasma membrane. This activation was essential for enhancing myocardial regeneration in Pex3-KO mice following plasmalogen supplementation.

Peroxisome has long been regarded as enigmatic subcellular components in eukaryotic cells, but recent research is shedding light on their mysteries. Diseases associated with peroxisomal metabolism, such as X-linked adrenoleukodystrophy, adult Refsum disease, or Zellweger syndrome, are generally caused by single-gene mutations in peroxisomal transporter or enzyme-encoding genes[60–62]. Previously, the understanding of cardiac structural and functional impairment remained limited due to the short lifespan of patients with Zellweger disease. However, patients with mild peroxisomal disorders often present with developmental abnormalities in the heart[63,64]. To gain insights into the physiological changes in mammalian tissues with peroxisomal defects, mouse models with gene deficiencies in Pex5, Pex2, Pex11α, Pex11β, Pex13, Pex7, Acox1, Mfp2, Gnpat, and Scp2 have been developed in recent years, but whether cardiac defects were present in this model has not been a primary research focus[65–68]. Therefore, our study based on the Pex3-KO mice myocardium provides novel insights into cardiac regenerative repair therapies, and establishes a valuable animal model to decipher whether peroxisomal functions primarily act as contributors or secondary modifiers in cardiac metabolic diseases. In addition to elucidating the role of peroxisome-related factors in the pathogenesis of diseases, exploring targeted therapies similar to those used for mitochondria holds excellent promise. Targeting peroxisome-related molecular functions could provide a ray of hope for precise treatments of related cardiac diseases. Therapeutic protein delivery for peroxisomal biosynthetic disorders is already underway[69]. Continuous advancements in research related to various peroxisomal functions, such as lipid metabolism and redox homeostasis, might steadily approach an improved understanding of pathogenesis and the development of therapies targeting these pathological conditions.

However, some limitations should be noted. We found experimentally that PEX3-related peroxisome function is activated after neonatal myocardial injury, whereas the expression is decreased after adult myocardial injury. So, we think activation of PEX3-related peroxisome function is closely related to the regenerative repair process after myocardial injury.

This functional mechanism is preserved in neonatal mice, but lost in adult mice. Peroxisomes, as the prominent effector organelles of reactive oxygen species, play an essential role in the cellular stress response and the regulation of cellular redox homeostasis, and peroxisomes are subjected to mobility regulation to adapt to changing cellular demands and different external environments[70]. In this study, we explored the underlying molecular mechanisms of PEX3, to see if PEX3 could act as a critical factor in the regulation of regenerative repair even after myocardial injury in adult mice. Deciphering the biological processes that dynamically control peroxisomes remains a significant challenge that we will pursue further in future work. Although Pex3-KO might largely affect the fate of CMs, the exact impacts and mechanisms of Pex3-KO or PEX3 overexpression on fibroblast, and the impact on non-CMs as well as the interaction on myocardial regeneration are not observed in this study and warrant further experimental studies. In addition, although we found that overexpression of PEX3 systemically can broadly promote CMs proliferation, whether the systemic administration of PEX3 has potential ectopic tumourigenicity is still an important work, which will be explored in subsequent studies.

In conclusion, we found that the peroxisome metabolic function is closely related to myocardial regenerative repair. PEX3, as an essential regulator of peroxisome-related functions, could promote myocardial regenerative repair through plasmalogen metabolism to facilitate plasma membrane localization of ITGB3, and activate the AKT/GSK3β signaling pathway. Our research highlights the indispensable role of PEX3 and ITGB3 in myocardial regenerative repair following injury, which may serve as novel therapeutic targets for the treatment of MI.

## Methods
### Animals
All animal experiments were performed in accordance with the Guide for the Care and Use of Laboratory Animals (NIH Publication No. 86-23, revised 2011) and were approved by the Animal Care and Use Committee of Nanjing Medical University (NO.1601038). Neonatal (P1) or adult (P56) wild-type C57BL/6 mice were purchased from the animal center of Nanjing Medical University and reared in a specific pathogen-free environment. Myh6-Cre mice and Pex3-floxed mice were transferred from the Jiahao Sha lab. Cardiomyocyte-specific Pex3 knockout mice were generated by crossbreeding between Myh6-Cre mice and Pex3-floxed mice. We have complied with all relevant ethical regulations for animal use.

In the study, we used Anaesthetic and analgesic agents as follows: 1.2% Avertin (Sigma, T48402, Germany) at a dose of 1ml/20g body weight by intraperitoneal injection to relieve pain/stress before MI surgery. After MI surgery, lab fellows inspected mice once a day for 28 days and registered them in the recording card. Carbon dioxide euthanasia is used for terminating mice. All anesthetic and analgesic agents used in the study were approved by the Animal Care and Use Committee of Nanjing Medical University.

### Neonatal mice AR model
The comparison of AR surgery involved the following groups: P1 Pex3-KO or WT mice. The AR models were created using a previously described method[71]. Briefly, the P1 mouse was anesthetized by

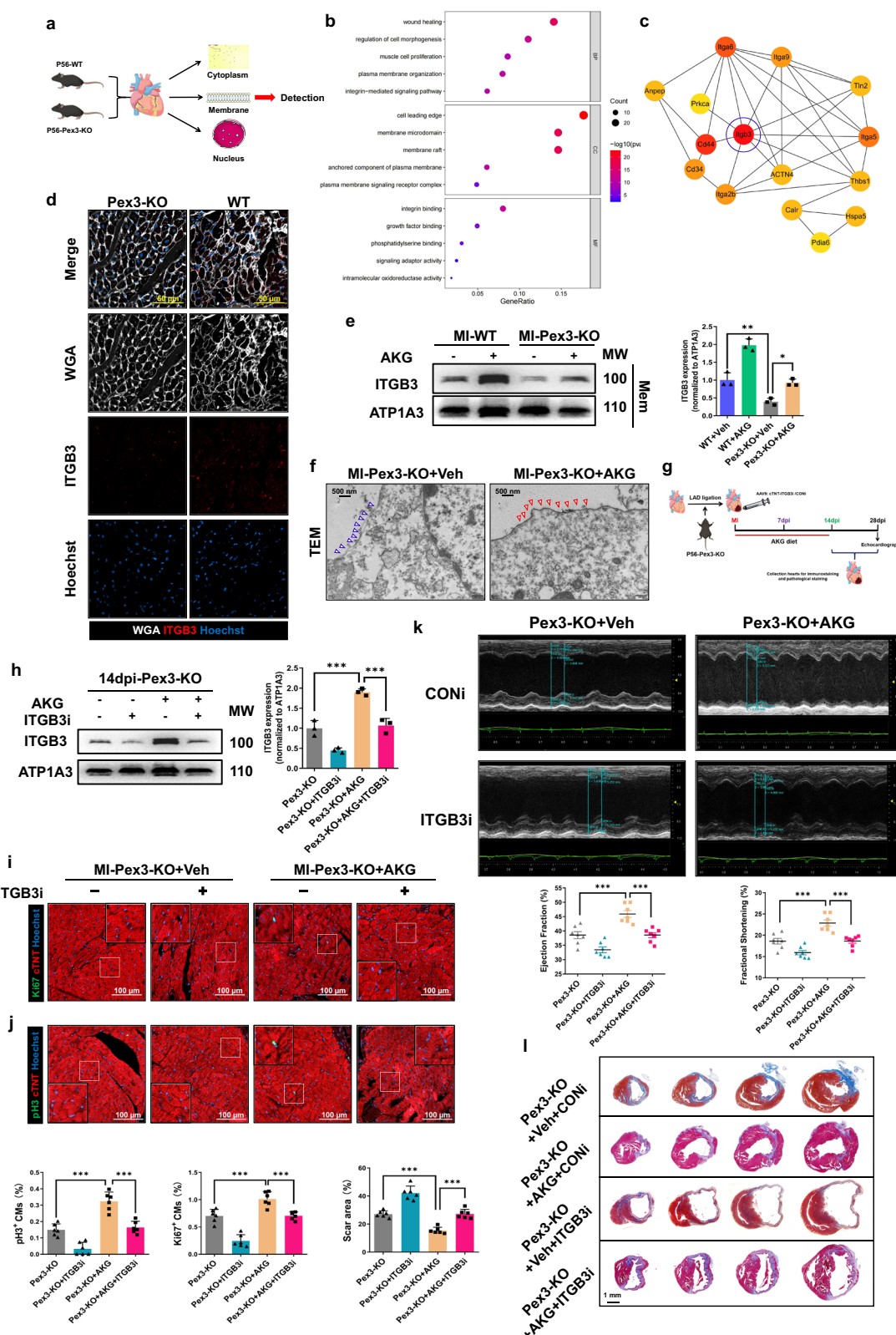

immersion in an ice bath for 3 min. Following the skin incision, the heart was exposed by carefully dissecting the intercostal muscles at the fourth intercostal space. Approximately one square millimeter of the apex was excised perpendicular to the long axis of the heart via ophthalmic scissors. Finally, the thoracic wall and skin were sutured using a 6–0 non-absorbable silk suture.

## Adult mice MI model

The comparison of MI surgery was conducted on the following groups: (1) P56 Pex3-KO or WT mice; (2) P56 WT mice injected with adeno-associated virus (AAV9: cTNT-PEX3/ITGB3/ITGB3i or AAV9: cTNT-CON/CONi). The MI model was created using a previously described method[71]. Briefly, P56 mouse was anesthetized by intraperitoneal injection of 1.2 % Avertin at

**Fig. 7 | PEX3 interferes with ITGB3 membrane localization through plasmalogen metabolism. a** Schematic illustration of the experimental design. **b** Gene ontology analysis of differentially down-expressed genes (Supplemental Fig. 10b) in WT and Pex3-KO adult mice heart. FC > 1.2, $P < 0.05$. **c** Using the Hubba plugin for Protein-Protein Interaction Networks to analyze the network topology of critical proteins in differentially expressed plasma membrane proteins. **d** Representative images of ITGB3 (red) expression in WT and Pex3-KO mice. WGA = White, Hoechst = Blue. Scale bar = 50 μm. **e** Western blotting and quantification analysis of ITGB3 in WT and Pex3-KO mice after MI treated with AKG for 2 weeks at 14 dpi ($n = 3$). **f.** Representative TEM images of the plasma membrane structure between Pex3-KO mice after MI and Pex3-KO mice after MI treated with AKG for 2 weeks at 14 dpi. Scale bar = 500 nm. **g** Schematic illustration of the experimental design. **h** Western blotting and quantification analysis of ITGB3 in Pex3-KO mice after MI and treated with AAV9-cTNT-ITGB3i (CONi) and AKG for 2 weeks at 14 dpi ($n = 3$). **i, j** Representative images and quantification analysis of Ki67 and pH3 (green) in Pex3-KO mice after MI and treated with AAV9-cTNT-ITGB3i (CONi) and AKG for 2 weeks at 14 dpi ($n = 6$). cTNT = Red, Hoechst = Blue. Scale bar = 100 μm. (Ki67: 3840 CMs in the Pex3-KO+CONi+Veh group, 3418 CMs in the Pex3-KO+ITGB3i +Veh group, 3795 CMs in the Pex3-KO+AKG+CONi group, 3491 CMs in the Pex3-KO+AKG+ITGB3i group; pH3: 3707 CMs in the Pex3-KO+CONi+Veh group, 4379 CMs in the Pex3-KO+ITGB3i+Veh group, 4894 CMs in the Pex3-KO +AKG+CONi group, 4035 CMs in the Pex3-KO+AKG+ITGB3i group). **k** Echocardiography measurements of ejection fraction and fractional shortening in Pex3-KO mice after MI and treated with AAV9-cTNT-ITGB3i (CONi) and AKG for 2 weeks at 28 dpi ($n = 7$). **l** Masson staining and quantification analysis of scar tissue in the injured heart of Pex3-KO mice after MI and treated with AAV9-cTNT-ITGB3i (CONi) and AKG for 2 weeks at 28 dpi ($n = 6$). Scale bar = 1 mm. Two-way ANOVA and Tukey's Multiple Comparison Test were performed for (**e, h–l**). Data shown as mean ± SEM. *$P < 0.05$, **$P < 0.01$, ***$P < 0.001$.

a dose of 1 ml/20 g body weight before the surgery. The fourth intercostal muscle was bluntly separated, and the MI model was induced by ligating the left anterior descending coronary artery using a 7–0 non-absorbable silk suture. Subsequently, the adeno-associated virus (AAV9: cTNT-PEX3/ ITGB3/ITGB3i or AAV9: cTNT-CON/CONi) was injected at three locations around the ligated myocardium using a 36G needle. The total amount of virus injected was $1.5 * 10^9$ viral genomes. The thoracic wall and skin were sutured using a 3–0 non-absorbable silk suture, and the mouse was placed on a heated stage until they recovered.

We constructed the MI model of adult mice and administered by tail vein at 3 dpi. The total amount of virus (AAV9: cTNT-PEX3 or AAV9: cTNT-CON) injected was $3*10^{11}$ PFU into each mouse.

### Study protocols

AR was performed in P1 mice and the detailed methods for anesthesia and surgery were included in the Methods. Mice were euthanized by $CO_2$ inhalation at indicated time point. PEX3 expression was verified in the resection border zone at 6 days post-resection (samples from the sham group were extracted from the same locations). EdU solution was intraperitoneally injected at 4 dpr (350 μg per mouse diluted in 70 μl PBS). Cell proliferation indicators (EdU+, Ki67+, pH3+, Aurora B+, the number of mono-, bi-, and multinucleated CMs, CMs size) in the border zone and the normal zone were determined by immunofluorescent staining at 6 dpr. Cardiac function was evaluated by echocardiography at 1 dpr and 28 dpr. Scar area and myocardial fibrosis were determined at 28 dpr by Masson staining.

MI and adeno-associated virus injection was performed in P56 mice and the detailed methods for anaesthesia and surgery were included in the Methods. Mice were euthanized by $CO_2$ inhalation at indicated time point. PEX3 expression was determined in the infarction border zone at 14 days post-infarction. EdU solution was intraperitoneally injected at 8, 10, and 12 dpi (500 μg per mouse diluted in 100 μl PBS). Cell proliferation indicators (EdU+, Ki67+, pH3+, Aurora B+, CMs size) in the border zone and the normal zone were determined by immunofluorescent staining at 14dpi. The proportion of mono-, bi-, and multinucleated CMs was evaluated after the isolation of CMs at 14 dpi. Cardiac function was evaluated by echocardiography at 28dpi. Scar area and myocardial fibrosis were determined at 7dpi and 28dpi by Masson staining.

Myocardial collection from WT and Pex3-KO mice was performed for morphological and functional experiments at P1, P3, P7, P14 and P28. AR or MI surgery was performed on P1 or P56 Pex3-KO and WT mice, followed by the abovementioned experiments.

To evaluate the potential role of plasmalogen in promoting CMs proliferation, we performed dietary interventions by administering AKG Mix (AKG Mix was prepared as previously described[28], batyl alcohol: chimyl alcohol: selachyl alcohol = 1:1:1) at 20 mg dissolved in 400 μL deionized water per mouse via oral gavage for consecutive 2 weeks in WT or Pex3-KO mice.

To determine if the AKT pathway plays a downstream role following the Pex3/plasmalogen/ITGB3 axis, we selected and interfered with two critical molecular targets, AKT and GSK-3β. Briefly, Neonatal mice cardiomyocytes (NMCMs) were treated with SC79[17] (15 μM, 8 h) to activate AKT, or TDZD8[72] (20 μM, 4 h) to inhibit GSK-3β. In vivo, SC79 (20 mg/kg) or TDZD8 (1 mg/kg,) was intraperitoneally injected after MI once a day in the first week and once every 2 days in the 2nd week.

### Echocardiography

Cardiac function was assessed by echocardiography (1/28dpr on AR mice, 1/7/28dpi on MI mice). For echocardiography, the adult mouse was anesthetized with 0.5–1.0% isoflurane and the neonatal mouse was conscious. Ejection fraction and fractional shortening were measured with a 40 MHz mouse ultrasound probe of Visual Sonics Vevo 2100 (VisualSonics, Toronto, Canada). All echocardiographic mice were performed in the animal center of Nanjing Medical University.

### Recombinant adenovirus and adeno-associated virus 9 (AAV9)

Recombinant adenovirus of scrambled shRNA for PEX3 (Ad5: cTNT-PEX3i) and control (Ad5: cTNT-CONi) were purchased from HANBIO company (Shanghai, China). The target sequence of shRNA for PEX3 was 5'-CGGAGAGTCTCACAGCTCTGTTGAA-3'. Adenovirus (Ad5: cTNT-PEX3) and AAV9 of PEX3 (AAV9: cTNT-PEX3) were constructed to enhance PEX3 expression. Ad5: cTNT-CON and AAV9: cTNT-CON were used as controls. Adenovirus and AAV9 vectors were also purchased from HANBIO company (Shanghai, China). Adenovirus and AAV9 of scrambled shRNA for ITGB3 (Ad5: cTNT-ITGB3i, AAV9: cTNT-ITGB3i, target sequence: 5'-GCTTGCCCATGTTTGGCTACA-3') was constructed to confer myocardium-targeted ITGB3 inhibition and relevant empty vectors were used as control. AAV9 of ITGB3 (AAV9: cTNT-ITGB3) were constructed to enhance ITGB3 expression, and AAV9: cTNT-CON were used as controls.

### NMCMs and non-CMs culture and intervention

NMCMs were isolated from 1 day-old C57BL/6J mice (50–100 mice each time) as previously described[71]. Briefly, the ventricular myocardium was collected and washed out after collection. Myocardium was then digested into single cells with 20 ml digestive solution containing 0.06 g/100 ml trypsin and 0.04 g/100 ml collagenase II for 6 min each time. After incubation with DMEM containing 10% FBS for 45 min, non-CMs were isolated through the differential attachment technique, and the suspension was further centrifuged (3000 rpm, 30 min) to remove fibroblasts with Percoll liquid. The NMCMs in the middle layer were then collected and cultured in an incubator with 5% $CO_2$ at 37 °C for another 48 h. Isolated non-CMs were cultured in DMEM supplemented with 10% FBS medium.

To explore the functional role of PEX3 on CMs, we incubated NMCMs with serum-free DMEM for 6 h and then transfected with adenovirus for another 6–8 h. Afterwards, NMCMs were set in a complete medium for 24–48 h.

For oxygen-glucose deprivation treatment, NMCMs were incubated with glucose and serum-free DMEM in AnaeroPACK Rectangular Jar

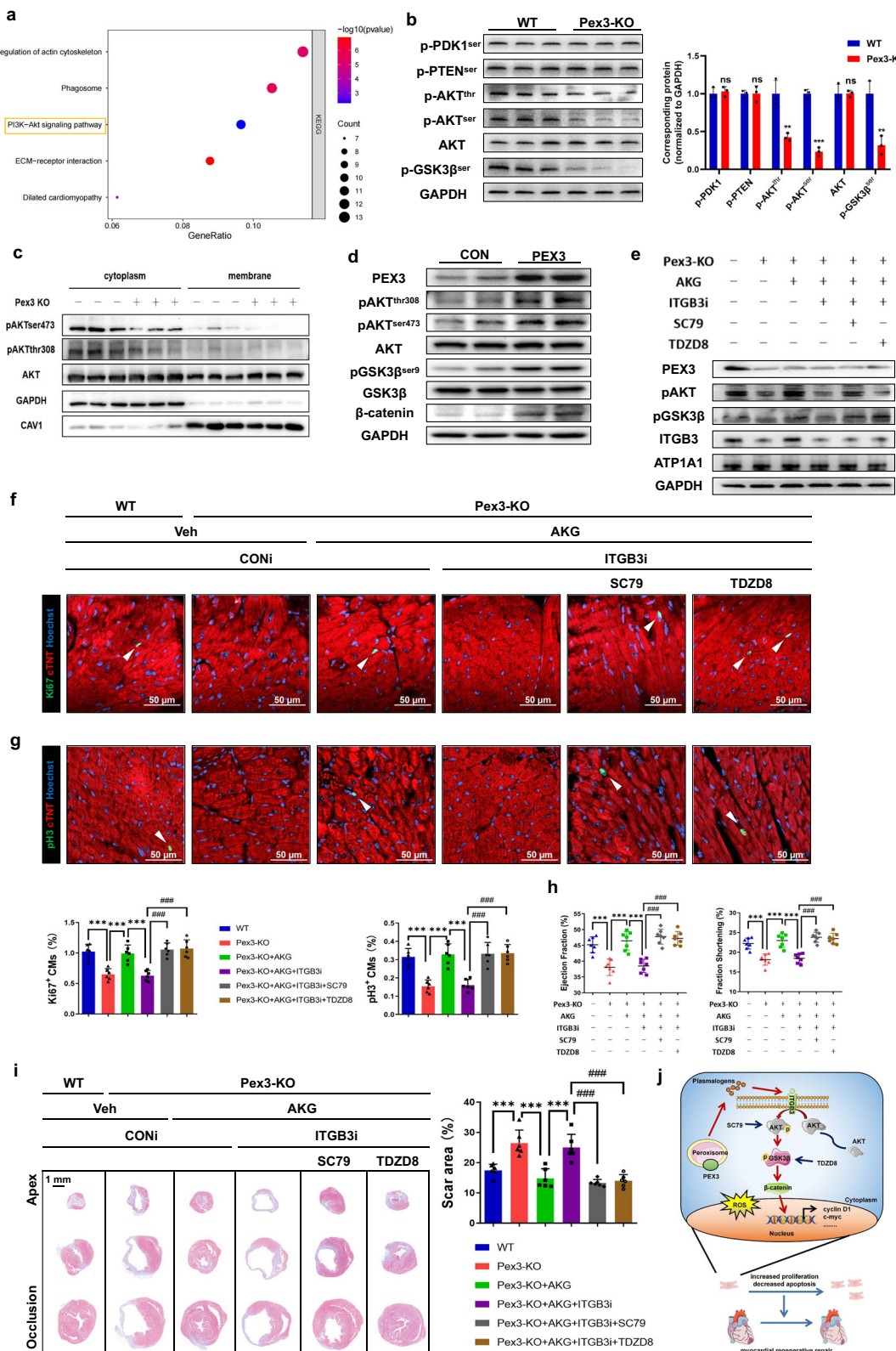

(Mitsubishi gas chemical company, INC, Japan) with 95% $N_2$ and 5% $CO_2$ at 37 °C for 8 h.

CMs viability was assessed by Cell Counting Kit-8 (CCK8, Beyotime, Shanghai) according to the manufacturer's protocols. Briefly, NMCMs were cultured in 96-well microplates at a density of 6000/well. After oxygen-glucose deprivation treatment and adenovirus transfection, 10 μL of CCK-8 reagent was added to each well and then cultured for another 2 h.

Absorbance at 450 nm was analyzed using a microplate spectrophotometer. All experiments were conducted in three duplicates.

### Juvenile and adult mice CMs isolation, culture, and intervention
C57BL/6J mice (male, 4 and 8 weeks old) were used to isolate primary adult CMs. First, the mice were anesthetized with 2% isoflurane and the chest was opened to reveal the descending aorta and inferior vena cava. 7 ml EDTA

**Fig. 8 | PEX3 regulates ITGB3-mediated AKT/GSK3β pathway to promote myocardial regenerative repair. a** Kyoto Encyclopedia of Genes and Genomes analysis of differential expression genes (Supplemental Fig. 4a) in WT and Pex3-KO mice myocardium. *P*-value < 0.05. **b** Western blotting and quantification analysis of p-PDK1$^{ser241}$, p-PTEN$^{ser380}$, p-AKT$^{thr308/ser473}$, AKT and p-GSK3β$^{ser9}$ in WT and Pex3-KO mice (*n* = 3). **c** Western blotting of AKT and p-AKT$^{thr308/ser473}$ in the cytoplasm and membrane of WT and Pex3-KO mice at 14dpi (*n* = 3). **d** Western blotting of PEX3, p-AKT$^{thr308/ser473}$, AKT, GSK3, p-GSK3β$^{ser9}$ and β-catenin in primary cardiomyocytes transfected with Ad5-cTNT-PEX3/CON (*n* = 4). **e** Western blotting of PEX3, p-AKT, p-GSK3β and ITGB3 in Pex3-KO mice after MI and treated with AAV9-cTNT-ITGB3i (CONi), AKG, SC79 or TDZD8 at 14dpi (*n* = 3). **f, g** Representative images and quantification analysis of Ki67 and pH3 (green) in Pex3-KO mice after MI treated with AAV9-cTNT-ITGB3i (CONi), AKG, SC79 or TDZD8 at 14 dpi (*n* = 6). cTNT = Red, Hoechst = Blue. Scale bar = 50 μm. (Ki67: 3669 CMs in the WT group, 3535 CMs in the Pex3-KO group, 3731 CMs in the Pex3-KO+AKG group, 3401 CMs in the Pex3-KO+AKG+ITGB3i group, 3416 CMs in the Pex3-KO+AKG+ITGB3i+SC79 group, 3721 CMs in the Pex3-KO+AKG+ITGB3i+TDZD8 group; pH3: 4821 CMs in the WT group, 4570 CMs in the Pex3-KO group, 4862 CMs in the Pex3-KO+AKG group, 4362 CMs in the Pex3-KO+AKG+ITGB3i group, 4953 CMs in the Pex3-KO+AKG+ITGB3i+SC79 group, 4879 CMs in the Pex3-KO+AKG+ITGB3i+TDZD8 group). **h** Echocardiography measurements of ejection fraction and fractional shortening in Pex3-KO mice after MI and treated with AAV9-cTNT-ITGB3i (CONi), AKG, SC79 or TDZD8 at 28 dpi (*n* = 7). **i** Masson staining and quantification of scar tissue in the injured heart of Pex3-KO mice after MI and treated with AAV9-cTNT-ITGB3i (CONi), AKG, SC79 or TDZD8 at 28 dpi (*n* = 6). Scale bar = 1 mm. **j** Schematic representation of how PEX3 promotes myocardial regenerative repair through plasmalogen metabolism to facilitate plasma membrane localization of ITGB3, activating the AKT/GSK3β signaling pathway. Unpaired *t*-test applied for (**b**). One-way ANOVA and Tukey's Multiple Comparison Test were performed for (**f–i**). Data shown as mean ± SEM. N.S, Not Significant, \*\**P* < 0.01, \*\*\**P* < 0.001. Pex3-KO + AKG + ITGB3i vs. Pex3-KO + AKG + ITGB3i + SC79/TDZD8, $^{###}P$ < 0.001.

---

buffer (pH 7.8) was manually perfused within 1 min to remove as much blood as possible. Next, the heart was perfused with 10 ml buffer with 5 mmol/l EDTA (around 6 min), 3 ml perfusion buffer (around 2 min), and 25–50 ml of collagenase buffer (20 min). The heart was then pulled apart into roughly 1 mm × 1 mm pieces using forceps. After complete digestion, 5 ml stop buffer was added to the cell suspension to inhibit enzyme activity, and the cells were filtered through a 100 μm cell strainer. Cells were collected by gravity settling for 20 min, and then resuspended in calcium reintroduction buffer three times to achieve healthy populations of calcium-tolerant cells. The cells were resuspended in a prewarmed plating medium and cultured at 37 °C in an incubation box with 5% CO$_2$ atmosphere. After 1 h, the culture medium was used to incubate CMs and was changed every 48 h. The culture medium was M199, supplemented with 5% Bovine serum albumin, 100 × ITS, 1 mol/L BDM, 100 × CD lipid, and 100 × penicillin/streptomycin solution.

For evaluation of mono-, bi-, and multinucleated CMs numbers, primary CMs from Pex3-KO and WT (MI+AAV9-cTNT-CON, MI+AAV9-cTNT-PEX3) mice (P28 sham and P56 MI at 14dpi) were isolated, cultured and stained with cTNT and Hoechst.

### RNA-sequencing and data analysis
Total RNA from Pex3-KO mice and WT mice was extracted using TRIzol reagent. The purity and quantity of the RNA were assessed using the NanoDrop 2000 spectrophotometer. A transcriptome library was constructed following the instructions of the VAHTS Universal V5 RNA-seq Library Prep kit. The library was then sequenced on the Illumina NovaSeq 6000 platform, generating 150 bp paired-end reads. The raw reads were processed using the fastp software, and alignment to the reference genome was performed using the HISAT2 software. To identify differentially expressed genes, differential gene expression analysis was performed using the DESeq2 software. Genes with a *q*-value < 0.05 and a fold change > 2 or fold change < 0.5 were considered as differentially expressed genes.

Gene Ontology analyses were performed using the "ClusterProfiler" package (v 4.2.2) in R (v 4.1.1). Biological process, cellular component and molecular function were included in Gene Ontology analysis. *P* < 0.05 was set as the cut-off criteria.

### Extraction of cardiac tissue cell membranes and proteomics analysis
The extraction of membrane proteins from Pex3-KO mice and WT mice myocardium was performed using the MinuteTM Plasma Membrane and Cytosol Fractionation Kit (Invent Biotechnologies, Inc., SM-005). Briefly, the procedure involved the following steps: (1) Fresh myocardium (30 mg) was placed on the column sleeve of a centrifuge tube. (2) 200 μl of Buffer A was added, and the tissue was ground for 1 min. (3) 300 μl of Buffer A was added to the centrifuge tube, and the mixture was incubated on ice for 5 min. (4) The tube was centrifuged at 16,000 g for 30 s, and the pellet obtained was

the total membrane fraction. (5) Next, 200 μl of Buffer B was added to resuspend the pellet, and the tube was incubated at 4 °C. (6) After 5 min of centrifugation at 7000 g, the supernatant was transferred. (7) 1.6 ml of pre-chilled PBS was added, mixed thoroughly, and centrifuged at 16,000g for 30 min. (8) The resulting pellet was the extracted plasma membrane fraction.

The obtained plasma membrane fractions from Pex3-KO mice and WT mice myocardium were subjected to mass spectrometry based proteomics analysis. The trypsin-digested peptides were desalted using an HLB column and then vacuum freeze-dried. The peptides were dissolved in 0.2 M TEAB and labeled according to the instructions of the Tandem Mass Tag reagent kit. The labeled peptides were subjected to high-pH reversed-phase HPLC separation using an XBridge BEH130 C18 column (300 μm × 150 mm, 1.7 μm, Waters). The peptide powder was reconstituted in 0.1% (v/v) formic acid and analyzed using the EASY-nLC 1200 ultra-high-performance liquid chromatography system. The eluted peptides were ionized and then analyzed using the Thermo Scientific Orbitrap Fusion™ Lumos™ Tribrid™ mass spectrometer. The second-level mass spectrometry data were searched using Proteome Discoverer (v2.4). For MS analysis, the peptides were reconstituted in 0.1% formic acid and analyzed using the LTQ Orbitrap Velos mass spectrometer (Thermo Fisher Scientific, San Jose, California) coupled to the Easy-nLC 1000 system. Mouse protein sequence searches were performed using the MaxQuant software (version 1.3.0.5) against the Universal Protein Resource database. False discovery rate cut-offs were set to 0.01 for proteins, peptides, and sites.

Gene ontology and Kyoto Encyclopedia of Genes and Genomes analyses were performed using the "ClusterProfiler" package (v 4.2.2) in R (v 4.1.1). Biological process, Cellular component and Molecular function were included in Gene ontology analysis. *P* < 0.05 was set as the cut-off criteria. The Search Tool for the Retrieval of Interacting Genes (STRING, version 11.0; https://string-db.org) database was used to predict the potential protein-protein interaction. The Protein-Protein Interaction network was constructed and visualized using Cytoscape software 3.10.0[73], and hub genes were identified by the Hubba plugin.

### Lipidomics and data analysis
Lipidomics analysis was performed on three groups of mice myocardium: WT-MI, Pex3-KO-MI, and Pex3-KO-MI+AKG. In brief, the Thermo Scientific™ Q Exactive™ Hybrid Quadrupole-Orbitrap Mass Spectrometer was used for fast, reliable identification, quantification, and confirmation of compounds. The LC-MS results were analyzed using Thermo Scientific™ LipidSearch™ software.

Data preprocessing was conducted before pattern recognition. The raw data were imported into LipidSearch software, and the search results for each individual sample were aligned within a specific retention time range. The results were then combined into a single report, generating a processed data matrix. The data matrix included sample information, identification

results for each peak, lipid class, fatty acid information, branch information, retention time, mass-to-charge ratio, molecular formula, and mass spectrometry response intensity (peak area), among other parameters.

For differential lipid compound selection, a $t$-test was performed to calculate $p$-values. Variables with a $p$-value < 0.05, combined with fold change values representing inter-group differences, were further screened as differentially expressed variables.

## Flow cytometry

NMCMs were pelleted and washed with PBS after pretreatment, followed by centrifuging at 1000 rpm for 10 min. While vortexing to loosen the pellet, 5 mL of cold 70–80% ethanol drop by drop into the NMCM pellet and then incubated at -20 °C for 2 h. The NMCMs were then washed with PBS and Stain Buffer and centrifuged at 1500 rpm for 10 min. For PI/RNase staining, the pellet was resuspended in 0.5 mL of PI/RNase Staining Buffer and incubated at room temperature for 15 min. Then, the cell suspension was ready for flow cytometry assay. Data was generated by BD FACSCalibur, CellQuest Pro (BD, USA).

## Immunofluorescent staining

Paraffin-embedded myocardium slices were subjected to deparaffinization and antigen retrieval. CMs were fixed in 4% PFA for 15 min. The samples were then permeabilized using 0.1% Triton X-100 in PBS and blocked with 5% bovine serum albumin for 2 h. Subsequently, the myocardium slices or CMs were incubated with primary antibodies overnight at 4 °C, followed by secondary antibody and Hoechst33342 incubation. The primary antibodies used in this study are as follows: cardiac troponin T (cTNT) (Abcam; ab8295), Vimentin (Proteintech, 10366-1-AP), CD68 (Proteintech, 28058-1-AP), CD31 (Proteintech, 11265-1-AP), DYKDDDDK (Proteintech, 20543-1-AP), α-SMA (Proteintech, 14395-1-AP), PMP70 (Abcam, ab109448), PEX3 (Invitrogen, PA5-115740), Ki67 (Abcam, ab16667), phosphorylated-histone 3 (pH3) (CST, 9701), Aurora B (Abcam, ab2254), γH2X (Abcam, ab81299), ITGB3 (Abcam, ab179473). WGA staining (1:200, w32466, Thermo) was used to label cell contours. Terminal deoxynucleotidyl transferase-mediated dUTP in situ nick end labeling (TUNEL, Vazyme. A113-01) staining and 5-ethynyl-2′-deoxyuridine (EDU, Invitrogen, C10637) staining was performed according to the manufacturer's protocols. The images were taken by confocal microscope (Zeiss, Oberkochen, Germany).

In vitro, isolated CMs were cultured in 24-well plates, treated with relevant reagents and fixed with 4% PFA for 15 min. After blocked with 10% goat serum for 2 h, proliferation indicators (EdU, Ki67, pH3, Aurora B) and Tunel staining were performed. The images were taken by confocal microscope (Zeiss, Oberkochen, Germany).

For ROS detection, frozen sections of fresh heart tissue were incubated with the DHE probe for 30 min at room temperature. Subsequently, Hoechst staining was performed for an additional 10 min to visualize the nuclei. DHE staining for primary CMs was performed without 4% PFA and 10% goat serum treatment. The images were captured using a confocal microscope (Zeiss, Oberkochen, Germany), and the levels of ROS were determined by analyzing the IF intensity.

## CMs counting method

In vitro, under the view of a confocal microscope (10 * 10 pm), 8–10 fields of view were randomly selected from each well of a 24-well plate to count the number of CMs as well as mono-, bi-, and multinucleated CMs. The CMs density was then calculated by expressing the number of CMs per mm². In vivo, considering that the size of CMs varies in mice of different ages, we used confocal views of 40 * 10 pm and 20 * 10 pm, respectively, to count CMs. To ensure the accuracy of the count and nucleus proportion, only cells that were clear and had the maximum cross-sectional area were calculated. In the neonatal mice models, we selected three sections from the apex to the base of the heart, with a 50 μm interval, and counted 20 fields of view at 40 x magnification in each section. In the adult mice models, we selected three sections from the apex to the base of the heart, with a 20 μm interval, and counted 20 fields of view, randomly chosen from the border zone and

remote non-infarct area, in each section. The border zone was defined as myocardial tissue within 3 fields of view at 20 * magnification from the injury edge, and the remote non-infarct area was defined as myocardial tissue located > 2 mm away from the border zone.

## Histological analysis and assessment of infarct size

For IHC staining, paraffin-embedded heart slices (6 days post-AR/MI and sham) were blocked with 5% bovine serum albumin, followed by incubation with primary antibodies against PEX3 or PMP70 (diluted 1:200) overnight at 4 °C. Slices were then washed with PBS and incubated with the secondary antibody for 2 h. Diaminobenzidine staining was then performed for histochemical visualization. Images were taken under a microscope. Quantitative analysis of signal intensity in IHC staining was performed using the Indica Labs software (USA, Halo v3.0.311.314).

At 28 days after AR or MI, mice were euthanized, and their hearts were collected. The hearts were fixed in 4% PFA and subsequently embedded in paraffin. For the AR group, longitudinal slices were prepared, while for the MI group, four transverse sections were obtained (with an interval of > 180 μm) from the apex to the ventricle. These sections were then subjected to Masson staining using standard procedures. The scar area was quantified using Image J software.

## MDA detection

The assessment of lipid peroxidation was via measuring MDA levels (Beyotime, S0131S). Briefly, mice myocardium was lysed and centrifuged (14,000 g, 4 °C, 30 min) to obtain the supernatant. The supernatant (0.1ml) was mixed with the working buffer at 100 °C for 15 min. The OD value was measured at 535nm.

## Transmission electron microscopy

Briefly, the myocardium was cut into pieces < 1 mm³, fixed in glutaraldehyde, dehydrated in graded acetone, and embedded in Epon-Araldite resin. Images were taken with a FEI Tecnai G2 Transmission Electron Microscope (Thermo Fisher Scientific, USA) at 80kV.

## RNA extraction and quantitative real-time polymerase chain reaction analysis

Total RNA was extracted from mice myocardium using TRIzol reagent (Thermo Fisher Scientific) under the manufacturer's instructions. RNA was then reverse transcribed to cDNA with the PrimerScript RT Master Mix kit (Takara Bio Lnc, Kusatsu, Japan). qRT-PCR analysis was performed using the SYBR Green (Vazyme Biotech, Nanjing, China; Q131-02) on a QuantStudio 7 detector (Thermo Fisher Scientific, Waltham, USA). 18S was used as an internal control. Primer sequences used in this study are listed in Supplemental Table 3.

## Protein extraction and western blot analysis

Myocardium or NMCMs were lysed with lysis buffer containing 0.5%EDTA, 0.1% protease inhibitor and 1% phosphatase inhibitor. The prepared protein was separated in SDS-PAGE gel and then transferred to a polyvinylidene fluoride membrane (Millipore). After blocking with 5% bovine serum albumin for 2 h at room temperature, the blots were incubated with primary antibodies at 4 °C overnight, then conjugated to second antibodies at room temperature for 2 h. The primary antibodies used in this study are as follows: GAPDH (Proteintech, 10494-1-AP, diluted 1:1000), PMP70 (Abcam, ab109448, diluted 1:1000), PEX12 (Proteintech, 27011-1-AP, diluted 1:1000), PEX3 (Proteintech, 10946-1-AP, diluted 1:1000), Histone H3 (Abcam, ab1791, diluted 1:1000), Caveolin-1 (Abcam, ab32577, diluted 1:1000), ATP1A3 (Proteintech, 25727-1-AP, diluted 1:1000), ITGB3 (Proteintech, 18309-1-AP, diluted 1:1000), AKT (CST, 4691S, diluted in 1:1000). p-AKT S473 (CST, 4060S, diluted 1:1000), p-AKT T308 (CST, 13038S, diluted 1:1000), p-PTEN S380(CST, 9551S, diluted 1:1000), p-PDK1 S241 (CST, 3438S, diluted 1:1000), β-catenin (Abcam, ab223075, diluted 1:1000) GSK-3β (Abcam, ab32391, diluted 1:1000), p-GSK-3β (CST, 5558S, diluted 1:1000). Quantification of band intensity was carried out with Image J software.

## Statistics and reproducibility

The data were presented as mean ± SEM and analyzed with GraphPad Prism 8.0 software (San Diego, CA, USA). All data sets in the article passed the normality distribution test via the Shapiro-Wilk test. The unpaired and 2-tailed *t*-test was used for two groups. When variance homogeneity is inconsistent, an unpaired *t*-test with Welch's correction was used instead. One-way or two-way ANOVA, and following Tukey's Multiple Comparison Test, were employed to assess variances in multi-group comparison when required by the experimental design, as indicated in figure legends. Survival rate analysis was performed using the Kaplan–Meier method with the Log Rank test. The data were presented as mean ± SEM and analyzed with GraphPad Prism 8.0 software (San Diego, CA, USA). Statistical significance was considered when $P \leq 0.05$ (All *P*-value statistics were in Supplemental Table 4–16). All experiments were conducted with a minimum of three biologically independent replicates.

## Reporting summary

Further information on research design is available in the Nature Portfolio Reporting Summary linked to this article.

## Data availability

The cited public RNA-seq data is from the GEO repository under the accession number GEO: GSE213233, and the spatial multi-omic data of human myocardial infarction is available at cellxgene (https://cellxgene.cziscience.com/collections/8191c283-0816-424b-9b61-c3e1d6258a77). The source data and the statistical data can be found in the Supplemental Data 1 and Supplemental Table 4–16. All data generated in this study are available from the corresponding author upon reasonable request.

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

## Acknowledgements

We thank for the technical assistance support from Jiangsu Province Collaborative Innovation Center for Cardiovascular Disease Translational Medicine. This work is supported by grants from the Key Clinical Frontier Technology Project of Department of Science and Technology of Jiangsu Provincial (NO. BE2022806), the National Natural Science Foundation of China Innovative Research Group Project (NO. 82121001), the National Natural Science Foundation of China (NO. 82070367), and a Project Funded by the Scientific Research Innovation Projects of Graduate Students in Jiangsu Province (No. SJCX21_0615). Figures 2a, 4a, 5a, 6c, 7a, 8j, Supplemental Fig. 3a, Supplemental Fig. 11a and Supplemental Fig. 12h were modified from Servier Medical Art (http://smart.servier.com/). Servier Medical Art is licensed under CC BY 4.0.

## Author contributions

J.S. conducted the in vitro and in vivo experiments, analyzed data and wrote the manuscript. Z.W. conducted the in vitro experiments and analyzed data. L.Z. conducted the in vivo experiments and analyzed data. T.Y. performed some of the in vivo experiments and wrote the manuscript. D.Z., S.W., L.G. T.S., and T.W. participated in some in vitro experiments. Y.B., J.C., H.W., and Q.W. performed some bioinformatic analysis. X.K., Y.J. and L.X. participated in designing the study. A.G., Y.Z. and F.C. participated in the statistical analysis and editing of the manuscript. L.W. and Y.C. (corresponding author) designed and supervised the study and performed manuscript editing.

## Competing interests

The authors declare no competing interests.
