## [Peer Review File · Communications Biology]

Reviewers' comments:

Reviewer #1 (Remarks to the Author):

The manuscript by Sun et al. aims to identify the role of the peroxisome regulator, PEX3, in myocardial regeneration. The authors observed an increase in RNA and protein levels of PEX3 during regeneration. They found that deletion of *pex3* reduces postnatal cardiomyocyte proliferation, while overexpression of *Pex3* promotes this proliferation in vitro. Importantly, they demonstrate that *Pex3* deletion impedes cardiomyocyte proliferation and neonatal heart regeneration following apical resection. Additionally, in vivo overexpression of *Pex3* via AAV9 enhances adult cardiomyocyte proliferation, decreases scar size, and improves cardiac function post-myocardial infarction (MI). To explore the mechanisms behind *Pex3*'s role in myocardial regeneration, the authors conducted lipidomic analyses and discovered a reduction in plasmalogen levels in *Pex3* knockout mice. Supplementation of plasmalogen improved the regenerative response in these mice. Further investigations revealed that plasmalogen regulates the localization of ITGB3 in the plasma membrane, which in turn mediates AKT signaling involved in the regenerative effects of *Pex3*. Therefore, the authors conclude that *Pex3* facilitates cardiomyocyte proliferation and regeneration through the modulation of plasmalogen metabolism, which affects ITGB3 and AKT signaling.

This is a comprehensive study that provides new insights into the role of *Pex3* and the peroxisome during heart regeneration. A few points can be addressed which would strengthen the conclusions of this study:

- Can the authors administer AAV9-cTnT-PEX3 systemically in adult mice at 3 days post-MI after confirmation of infarct sizes to fully establish its impact on regeneration. Additionally, the systemic injection might promote higher levels of cardiomyocyte proliferation throughout the myocardium rather than the border zone only.
- For plasmalogen supplementation in Figure 6 and 7, confirmation of infarct sizes and cardiac function at an earlier timepoint post-MI to ensure equal infarction is necessary, as it is difficult to interpret this data as regeneration rather than protection or surgical variability. While the current adult data is promising, it doesn't distinguish between protection or regeneration.

Reviewer #2 (Remarks to the Author):

The authors present a large set of experiments and data to help describe how the peroxisome is involved during regeneration. This has so far not been well understood and the manuscript adds an interesting layer to the complicated yet coordinated events following heart injury and will be of interest to the wider scientific community. However there are a number of points the reviewer would like to be experimentally addressed. Furthermore the text is dense, there is a lot of main figures and the ordering of both the main and supplemental figures requires the reader to jump

between figures making it difficult to follow the story line. Some editing (rearranging figures, moving main figures to supplement or removing entirely) will be beneficial for helping the reader follow the manuscript.

Major

-while the increase in PMP70 and PEX3 at the BZ in neonates supports the hypothesis, the decrease in PMP70 and PEX3 in adult IZ and BZ is confounding. Wouldn't the authors predict it to be unchanged? Does this suggest that in the adult context, there is a different feedback in peroxisome regulation when compared to the neonates and if so how is this regulated?

-the author states an increase in PMP70 in the immunostainings in supplemental Figure 1E. Please quantify this as it appears there are also PMP70 positive cells in non-cardiomyocytes that might be masking the actual result. The same for the non-cardiomyocyte PEX3 stains in Fig1E-F

-figure 1A shows increase in PEX3 but also PEX12. PEX12 is not mentioned at all, how does this fit in with the author's data?

-Ad5 PEX3 overexpression suggests it is cardioprotective, while AAV9 PEX3 overexpression suggests it promotes positive regeneration. Are these linked?

-I assume AR in line 247 means apical resection? I find it strange that this was the choice to induce heart injury. The neonatal LAD model (which is later used in the manuscript in adults) would be more appropriate to conduct the experiments since this model better induces the hypoxia/ischemia conditions seen following MI.

-increased fibrosis in Pex3-KO mentioned from line 263 onwards needs to be explored more. How is PEX3 and to a broader degree peroxisomes mechanistically affecting fibrosis? Are the fibroblasts being activated? How do peroxisomes interact with fibroblasts when the model presented is a cardiomyocyte specific KO? Is the increase in fibrosis coming from increased immune cell activity/activation? How this links to the proliferation data is also not clear

-similarly, in the PEX3 overexpression in adult MI model, the data suggests the scar is being removed. Is this really the case? If so how is this achieved given that AAV9 delivery is generally cardiomyocyte specific. Are other cell types such as fibroblasts or immune cells involved?

-echo analysis to assess function at 28dpi is line 294 is rather early to state a functional recovery, particularly when the scar is still present. Please provide later timepoints if the authors want to keep this statement or tone down

-it is unclear if the authors are trying to establish a direct link between proliferation and fibrosis following PEX3 modulation. The text in some places suggests it is while not in other places. Please be clearer and if a direct link is suggested, please provide the data to support these claims.

-ITGB3 appears to be downstream of PEX3, what do the peroxisomes look like in ITGB3 overexpression/knockdown? Is the PEX3/ITGB3 interaction direct or only indirect via plasmalogens?

-is peroxisome density (eg. PMP70 levels) modulated when ITGB3 is manipulated?

Minor

-the use of abbreviations and not defining what it actually is makes it difficult to follow. For example dps, dpr, AR, DCF, MOI are examples of abbreviations used but I actually have no idea what the

authors are referring to

- Please specify more clearly what is P7/6dps, P7/6dpr, P56/6dps/P56/6dpi. These abbreviations were not defined anywhere in the main text and i had to go hunting in an unrelated figure legend to figure out what the authors were talking about. Please rename these abbreviations to something that is more easy for the reader to understand. There is already so much abbreviations for other names in the text that the author's message gets lost
- why is P7/6dpr and P56/6dpi in Sup. fig1E and P7/6dpr and P56/6dpi in Sup. Fig 1F are being compared where elsewhere the same ages (P7 and P56) are compared. My confusion is again with these nomenclature, it is just too confusing to first figure out what the group is and why particular groups are being compared to
- supplemental table 1: are these annotations the authors made or based on literature? if based on literature, please cite
- does weak western blot PEX3 bands in Pex3-KO in sup. Fig 2C mean an incomplete knockout?
- can the authors explain why a lower percentage of mono nucleated CMs in Pex3-KO is expected? Isn't the increase in bi-nucleation an indicator of more mature CMs?
- why was the timepoint P56 not used for Pex3-KO experiments? Inclusion of this timepoint in the analysis for proliferation would help link it to figure 1
- It is not clear what age the cardiomyocytes are for the PEX3i experiments and how they relate to the KO data
- im confused in line 206 where there authors mention a knockdown strategy yet the preceding text from line 209 onwards talks about over expression
- can the authors provide data that Ad5 transfection is cardiomyocyte specific?
- can the authors provide data that the cardiomyocyte populations used for Ad5 transfection are pure and non-cardiomyocytes obtained from the isolations are not affecting the downstream analysis?
- can the authors provide data that the AAV9 transfections is targeting only cardiomyocytes?

Responses to the editorial office and reviewers:

Thanks for your advice and we are delighted to have the opportunity to revise our manuscript entitled "*PEX3 promotes regenerative repair after myocardial injury through facilitating plasma membrane localization of ITGB3*" (Manuscript Number: COMMSBIO-24-0061-T). We are grateful for the thoughtful and helpful comments from the editor and the reviewers on our manuscript. We carefully modified the manuscripts according to the comments. All changes in the manuscript are marked in red font in the revised Manuscript and Supplementary Materials. Below are our point-to-point responses.

Thanks again for your consideration.

Yours sincerely,
Lian-sheng Wang, PhD, MD.

Reviewer 1:

The manuscript by Sun et al. aims to identify the role of the peroxisome regulator, PEX3, in myocardial regeneration. The authors observed an increase in RNA and protein levels of PEX3 during regeneration. They found that deletion of *pex3* reduces postnatal cardiomyocyte proliferation, while overexpression of Pex3 promotes this proliferation in vitro. Importantly, they demonstrate that Pex3 deletion impedes cardiomyocyte proliferation and neonatal heart regeneration following apical resection. Additionally, in vivo overexpression of Pex3 via AAV9 enhances adult cardiomyocyte proliferation, decreases scar size, and improves cardiac function post-myocardial infarction (MI). To explore the mechanisms behind Pex3's role in myocardial regeneration, the authors conducted lipidomic analyses and discovered a reduction in plasmalogen levels in Pex3 knockout mice. Supplementation of plasmalogen improved the regenerative response in these mice. Further investigations revealed that plasmalogen regulates the localization of ITGB3 in the plasma membrane, which in turn mediates AKT signaling involved in the regenerative effects of Pex3. Therefore, the authors conclude that Pex3 facilitates cardiomyocyte proliferation and regeneration through the modulation of plasmalogen metabolism, which affects ITGB3 and AKT signaling.

This is a comprehensive study that provides new insights into the role of Pex3 and the peroxisome during heart regeneration. A few points can be addressed which would strengthen the conclusions of this study.

1. Can the authors administer AAV9-cTnT-PEX3 systemically in adult mice at 3 days post-MI after confirmation of infarct sizes to fully establish its impact on regeneration. Additionally, the systemic injection might promote higher levels of cardiomyocyte proliferation throughout the myocardium rather than the border zone only.

Response: Thanks so much for this important comment. We followed the suggestion and performed additional experiments. We injected AAV9-cTnT-PEX3 via the tail vein into adult mice with comparable LVEF at 3 days after myocardial infarction. We examined the proliferation levels

of cardiomyocytes in the infarct border zone as well as in the normal zone, respectively. The results showed that overexpression of PEX3 systemically promotes the proliferation of cardiomyocytes in the normal zone and border zone (**Supplementary Figure 7C-E**). However, whether systemic overexpression of PEX3 has potential ectopic tumourigenicity is still an important work, which will be explored in subsequent studies. The related contents are added in Methods (**Page 8, line 198-200**), Results (**Page 26, line 715-722**) and Discussion (**Page 36, line 1008-1011**) now.

2. For plasmalogen supplementation in Figure 6 and 7, confirmation of infarct sizes and cardiac function at an earlier timepoint post-MI to ensure equal infarction is necessary, as it is difficult to interpret this data as regeneration rather than protection or surgical variability. While the current adult data is promising, it doesn't distinguish between protection or regeneration.

Response: Thanks for the helpful comments. To ensure acceptable homogeneity of the infarction model in experimental animals, we included animals with comparable echocardiography-derived LVEF (**Supplementary Figure 9A** LVEF: $53.52\% \pm 0.8042$; **Supplementary Figure 11E** LVEF: $53.51\% \pm 0.7707$) at 1dpi in our study (**Supplementary Figure 9A** and **Supplementary Figure 11E**). Protection and repair after the myocardial injury is an ambitious scope that includes myocardial regeneration, inflammation regulation, neovascularisation and interstitial remodelling¹. In our work, we found that PEX3 mainly promotes cardiomyocyte proliferation to achieve regenerative repair after injury by detecting cardiomyocyte proliferation indexes. We have revised the expression accordingly in the manuscript.

1. Uygur A, Lee RT. Mechanisms of Cardiac Regeneration. Dev Cell. 2016 Feb 22;36(4):362-74. PMID: 26906733; PMCID: PMC4768311.

Reviewer 2:

The authors present a large set of experiments and data to help describe how the peroxisome is involved during regeneration. This has so far not been well understood and the manuscript adds an interesting layer to the complicated yet coordinated events following heart injury and will be of interest to the wider scientific community. However there are a number of points the reviewer would like to be experimentally addressed. Furthermore the text is dense, there is a lot of main figures and the ordering of both the main and supplemental figures requires the reader to jump between figures making it difficult to follow the story line. Some editing (rearranging figures, moving main figures to supplement or removing entirely) will be beneficial for helping the reader follow the manuscript.

Response: Thank you for the helpful comments. In the revised manuscript, we carefully considered all the related issues and made corresponding adjustments, modifications, and supplements to the figures and text.

Major

1. while the increase in PMP70 and PEX3 at the BZ in neonates supports the hypothesis, the decrease in PMP70 and PEX3 in adult IZ and BZ is confounding. Wouldn't the authors predict it to be unchanged? Does this suggest that in the adult context, there is a different feedback in peroxisome regulation when compared to the neonates and if so how is this

regulated?

Response: Thank you for the thoughtful comments. We found experimentally that PEX3-related peroxisome function is activated after neonatal myocardial injury, whereas the expression is decreased after adult myocardial injury. So, we think activation of PEX3-related peroxisome function is closely related to the regenerative repair process after myocardial injury. This functional mechanism is preserved in neonatal mice, but lost in adult mice. Peroxisomes, as the prominent effector organelles of reactive oxygen species, play an essential role in the cellular stress response and the regulation of cellular redox homeostasis, and peroxisomes are subjected to mobility regulation to adapt to changing cellular demands and different external environments². In this study, we explored the underlying molecular mechanisms of PEX3, to see if PEX3 could act as a critical factor in the regulation of regenerative repair even after myocardial injury in adult mice. Deciphering the biological processes that dynamically control peroxisomes remains a significant challenge that we will pursue further in future work. The above information is added in the revised manuscript. It reads (Page 36, line 990-1003): "However, some limitations should be noted. We found experimentally that PEX3-related peroxisome function is activated after neonatal myocardial injury, whereas the expression is decreased after adult myocardial injury. So, we think activation of PEX3-related peroxisome function is closely related to the regenerative repair process after myocardial injury. This functional mechanism is preserved in neonatal mice, but lost in adult mice. Peroxisomes, as the prominent effector organelles of reactive oxygen species, play an essential role in the cellular stress response and the regulation of cellular redox homeostasis, and peroxisomes are subjected to mobility regulation to adapt to changing cellular demands and different external environments¹. In this study, we explored the underlying molecular mechanisms of PEX3, to see if PEX3 could act as a critical factor in the regulation of regenerative repair even after myocardial injury in adult mice. Deciphering the biological processes that dynamically control peroxisomes remains a significant challenge that we will pursue further in future work".

1. Di Cara F, Savary S, Kovacs WJ, Kim P, Rachubinski RA. The peroxisome: an up-and-coming organelle in immunometabolism. *Trends Cell Biol.* 2023 Jan;33(1):70-86. Epub 2022 Jul 1. PMID: 35788297.

2. the author states an increase in PMP70 in the immunostainings in supplemental Figure 1E. Please quantify this as it appears there are also PMP70 positive cells in non-cardiomyocytes that might be masking the actual result. The same for the non-cardiomyocyte PEX3 stains in Fig1E-F

Response: Thanks for pointing out this important issue. As suggested, we quantified the intensity of PMP70 and PEX3 and found that PMP70 or PEX3 was overexpressed after neonatal myocardial injury and decreased after adult myocardial injury (**Supplementary Figure 1E and Supplementary Figure 2C**). Additional co-localization immunofluorescence experiments were performed, and results showed that PMP70 or PEX3 expression was not enriched in non-cardiomyocytes marked by Vimentin, CD68, and CD31 at P7/6dpr and P56/6dpi (**Supplementary Figure 1A-B and Supplementary Figure 2A-B**).

A. Representative images of PMP70 (red) at P7/6dpr of mice heart. cTNT (Cardiomyocyte) /Vimentin (Fibroblast) /CD68 (Macrophage) /CD31 (Endothelial cell) =Green, Hoechst=Blue. Scale bar=20 μ m. **B.** Representative images of PMP70 (red) at P56/6dpi of mice heart. cTNT (Cardiomyocyte) /Vimentin (Fibroblast) /CD68 (Macrophage) /CD31 (Endothelial cell) =Green, Hoechst=Blue. Scale bar=20 μ m.

A. Representative images of PEX3 (red) at P7/6dpr of mice heart. Vimentin (Fibroblast) /CD68 (Macrophage) /CD31 (Endothelial cell) =Green, Hoechst=Blue. Scale bar=20 μ m. **B.** Representative images of PEX3 (red) at P56/6dpi of mice heart. Vimentin (Fibroblast) /CD68 (Macrophage) /CD31 (Endothelial cell) =Green, Hoechst=Blue. Scale bar=20 μ m.

3. figure 1A shows increase in PEX3 but also PEX12. PEX12 is not mentioned at all, how does this fit in with the author's data?

Response: Thanks for the helpful comments. Yes, the expressions of PEX3 and PEX12 were both increased after neonatal myocardial injury, and the increase of PEX3 was more significant (**Figure 1B**). It is known that the most severe peroxisome-deficient phenotype was in yeast PEX3 cells, in which most of the peroxisome matrix and membrane proteins were mislocalised⁴. Whereas PEX12 is a part of genes that encode peroxins involved in matrix protein import. Mutation in PEX12 results in smaller, spherical peroxisomal structures⁵. In polymorphic H.PEX3 cells, only a fraction of the PMPs is present on peroxisomal membrane structures. These proteins are stable relative to other PMPs, which are also incorrectly localized to the cytoplasm⁶. As a result, the different peroxins acting on matrix protein import and peroxisomal membrane biogenesis are often unable to be aligned, and this may behave differently in different model organisms⁷. In general, PEX3 is a prerequisite for peroxisomal regulation and served as a key signaling in this study, the next, we will further explore whether PEX3 is valuable in myocardial regeneration. The above information has been added to the discussion section. It reads (Page 32-33, line 891-904): "We initially found that the expressions of PEX3 and PEX12 were both increased after neonatal myocardial injury, and the increase of PEX3 was more significant. It is known that the most severe peroxisome-deficient phenotype was in yeast PEX3 cells, in which most of the peroxisome matrix and membrane proteins were mislocalized¹. Whereas PEX12 is a part of genes that encode peroxins involved in matrix protein import. Mutation in PEX12 results in smaller, spherical peroxisomal structures². In polymorphic H.PEX3 cells, only a fraction of the PMPs is present on peroxisomal membrane structures. These proteins are stable relative to other PMPs, which are also incorrectly localized to the cytoplasm³. As a result, the different peroxins acting on matrix protein import and peroxisomal membrane biogenesis are often unable to be aligned, and this may behave differently in different model organisms⁴. In general, PEX3 is a prerequisite for peroxisomal regulation and served as a critical signaling in this study. Next, we further explored whether PEX3 is valuable in myocardial regeneration".

1. Shimozawa N, Suzuki Y, Zhang Z, Imamura A, Ghaedi K, Fujiki Y, Kondo N. Identification of PEX3 as the gene mutated in a Zellweger syndrome patient lacking peroxisomal remnant structures. *Hum Mol Genet.* 2000 Aug 12;9(13):1995-9. PMID: 10942428.
2. Yuan W, Veenhuis M, van der Klei IJ. The birth of yeast peroxisomes. *Biochim Biophys Acta.* 2016 May;1863(5):902-10. Epub 2015 Sep 11. PMID: 26367802.
3. Hazra PP, Suriapranata I, Snyder WB, Subramani S. Peroxisome remnants in pex3delta cells and the requirement of Pex3p for interactions between the peroxisomal docking and translocation subcomplexes. *Traffic.* 2002 Aug;3(8):560-74. PMID: 12121419.
4. Smith JJ, Aitchison JD. Peroxisomes take shape. *Nat Rev Mol Cell Biol.* 2013 Dec;14(12):803-17. PMID: 24263361.

4. Ad5 PEX3 overexpression suggests it is cardioprotective, while AAV9 PEX3 overexpression suggests it promotes positive regeneration. Are these linked?

Response: Thanks for the valuable comments. We apologize for the lack of a uniform description in our manuscript. Cardiac protection and repair after an injury is a grand scope, including cardiomyocyte proliferation, inflammation regulation, neovascularisation and interstitial

remodelling¹. In our research, transfection of Ad5: cTNT-PEX3 could promote the proliferation of neonatal mice primary cardiomyocytes in vitro. In vivo, transfection of AAV9: CTNT-PEX3 could promote the proliferation of cardiomyocytes and improve cardiac function after MI. The above results indicated that PEX3 promoted cardiomyocyte proliferation and cardiac repair. We have revised the expression accordingly in the manuscript.

1. Uygur A, Lee RT. Mechanisms of Cardiac Regeneration. *Dev Cell*. 2016 Feb 22;36(4):362-74. PMID: 26906733; PMCID: PMC4768311.

5. I assume AR in line 247 means apical resection? I find it strange that this was the choice to induce heart injury. The neonatal LAD model (which is later used in the manuscript in adults) would be more appropriate to conduct the experiments since this model better induces the hypoxia/ischemia conditions seen following MI.

Response: Thanks. AR refers to apical resection, which is a common animal model for studying myocardial regeneration in neonatal mice¹⁻³. LAD ligation of neonatal mouse hearts assisted by somatoscopy requires a very high surgical technique, and this operation is not yet established in our laboratory; the next, we will try this model in future studies. Thank you for this stimulating and helpful comment.

1. Ji X, Chen Z, Wang Q, Li B, Wei Y, Li Y, Lin J, Cheng W, Guo Y, Wu S, Mao L, Xiang Y, Lan T, Gu S, Wei M, Zhang JZ, Jiang L, Wang J, Xu J, Cao N. Sphingolipid metabolism controls mammalian heart regeneration. *Cell Metab*. 2024 Apr 2;36(4):839-856.e8. Epub 2024 Feb 16. PMID: 38367623.

2. Li Y, Feng J, Song S, Li H, Yang H, Zhou B, Li Y, Yue Z, Lian H, Liu L, Hu S, Nie Y. gp130 Controls Cardiomyocyte Proliferation and Heart Regeneration. *Circulation*. 2020 Sep 8;142(10):967-982. Epub 2020 Jun 30. PMID: 32600062.

3. Feng J, Li Y, Li Y, Yin Q, Li H, Li J, Zhou B, Meng J, Lian H, Wu M, Li Y, Dou K, Song W, Lu B, Liu L, Hu S, Nie Y. Versican Promotes Cardiomyocyte Proliferation and Cardiac Repair. *Circulation*. 2024 Mar 26;149(13):1004-1015. Epub 2023 Oct 27. PMID: 37886839.

6. increased fibrosis in Pex3-KO mentioned from line 263 onwards needs to be explored more. How is PEX3 and to a broader degree peroxisomes mechanistically affecting fibrosis? Are the fibroblasts being activated? How do peroxisomes interact with fibroblasts when the model presented is a cardiomyocyte specific KO? Is the increase in fibrosis coming from increased immune cell activity/activation? How this links to the proliferation data is also not clear

Response: Thanks for the insightful comments. As suggested, we performed immunofluorescence co-localization experiments of PEX3 and non-cardiomyocytes, and the results showed that PEX3 was mainly activated/inhibited in cardiomyocytes after neonatal/adult myocardial injury, but not in fibroblasts/endothelial cells/macrophages (**Figure 1E-F and Supplementary Figure 2A-B**). Therefore, we believe that when Pex3-KO mostly affects cardiomyocytes, future studies are needed to explore the above issued comments. We added the related study limitation now. It reads (**Page 36, line 1003-1008**): "Although Pex3-KO might largely affect the fate of CMs, the exact impacts and mechanisms of Pex3-KO or PEX3 overexpression, on fibroblast, the role of non-CMs including immune cells on fibrosis, their impact on regenerative capacity, the impact on non-CMs

as well as the interaction on fibrosis, CM proliferation as well as myocardial regeneration are not observed in this study and warrant further experimental studies".

7. similarly, in the PEX3 overexpression in adult MI model, the data suggests the scar is being removed. Is this really the case? If so how is this achieved given that AAV9 delivery is generally cardiomyocyte specific. Are other cell types such as fibroblasts or immune cells involved?

Response: Thanks for the fair comments. This is an assay of great value in previous studies in the field of cardiomyocyte proliferation as well as myocardial regeneration¹⁻². In our study, we focused on the impact of overexpression of PEX3 on proliferative replenishment of cardiomyocytes and their role in the reduction of fibrosis after myocardial injury, while their impact on regenerative capacity, the impact on non-cardiomyocytes as well as the interaction on fibrosis, cardiomyocyte proliferation as well as myocardial regeneration remains unknown and need to be verified in future studies. This study limitation is added in the revised manuscript now (Page 36, line 1003-1008).

1. Bae J, Salamon RJ, Brandt EB, Paltzer WG, Zhang Z, Britt EC, Hacker TA, Fan J, Mahmoud AI. Malonate Promotes Adult Cardiomyocyte Proliferation and Heart Regeneration. *Circulation*. 2021 May 18;143(20):1973-1986. Epub 2021 Mar 5. PMID: 33666092; PMCID: PMC8131241.
2. Wang Y, Li Y, Feng J, Liu W, Li Y, Liu J, Yin Q, Lian H, Liu L, Nie Y. Myd88 promotes Cardiomyocyte proliferation and Neonatal Heart regeneration. *Theranostics*. 2020 Jul 11;10(20):9100-9112. PMID: 32802181; PMCID: PMC7415811.

8. echo analysis to assess function at 28dpi is line 294 is rather early to state a functional recovery, particularly when the scar is still present. Please provide later timepoints if the authors want to keep this statement or tone down

Response: Thank you very much for your professional advice. We have modified the presentation of the 28dpi echocardiography results in the appropriate section of the manuscript to achieve more accuracy. It reads (Page 26, line 709-710): "echocardiography results revealed that PEX3 overexpression improved cardiac function at 28dpi".

9. it is unclear if the authors are trying to establish a direct link between proliferation and fibrosis following PEX3 modulation. The text in some places suggests it is while not in other places. Please be clearer and if a direct link is suggested, please provide the data to support these claims.

Response: Thanks for this comment. We apologize for the unclear presentation. Our results showed that modulating PEX3 expression could affect the proliferative capacity of cardiomyocytes after myocardial injury, which might lead to differences in regenerative repair or scar filling after myocardial injury. A solid direct link could not be established based on available data, since other factors might also contribute to the proliferation and fibrosis changes following PEX3 modulation. We have modified the manuscript accordingly. It reads (Page 25, line 679-680): "Moreover, Masson staining showed that PEX3 knockdown inhibited myocardial regeneration"; (Page 26, line 710-712): "Additionally, Masson staining demonstrated that PEX3 overexpression significantly reduced the infarct size"; (Page 27, line 733-734): "PEX3 knockdown significantly

increased the infarct size at 28dpi"; (Page 28, line 771-772): "In addition, Masson staining showed that plasmalogen supplementation could reduce the infarct size at 28dpi".

10. ITGB3 appears to be downstream of PEX3, what do the peroxisomes look like in ITGB3 overexpression/knockdown? Is the PEX3/ITGB3 interaction direct or only indirect via plasmalogens

Response: Thanks for the important comments. Yes, ITGB3 is a downstream regulatory target of PEX3. Our experiments in Result 8 (**Figure 7 and Supplementary Figure 10-11**) showed that PEX3 regulates the plasma membrane localization of ITGB3 and functions as a signal transducer by affecting acetylated phospholipid metabolism, thus regulating ITGB3. To further clarify that ITGB3 is a downstream regulatory target of PEX3, we further examined the changes in the expression of PEX3 and PMP70 after intervening with ITGB3, and the results showed that the alteration of ITGB3 did not affect the expression of PEX3 and PMP70 (**Supplementary Figure 10C-F**). The text has now been rephrased to present this point more clearly in the revised manuscript.

11. is peroxisome density (eg. PMP70 levels) modulated when ITGB3 is manipulated?

Response: Thanks for the comment. As suggested, we detected the expression changes of PMP70 after intervening in ITGB3, and the results showed that intervention in ITGB3 did not affect the expression of PEX3 and PMP70 (**Supplementary Figure 10D-F**).

Minor:

1. the use of abbreviations and not defining what it actually is makes it difficult to follow. For example dps, dpr, AR, DCF, MOI are examples of abbreviations used but I actually have no idea what the authors are referring to

Response: Thank you for bringing up the issue of unexplained abbreviations. In our manuscript, "dps" stands for days post sham, "dpr" stands for days post apical resection, "AR" stands for apical resection, "DCF" stands for 2',7'-Dichlorofluorescein, and "MOI" stands for multiplicity of infection. We have compiled a list of all the abbreviations used in the study now (Page 4-5, line 59-115).

2. Please specify more clearly what is P7/6dps, P7/6dpr, P56/6dps/P56/6dpi. These abbreviations were not defined anywhere in the main text and i had to go hunting in an unrelated figure legend to figure out what the authors were talking about. Please rename these abbreviations to something that is more easy for the reader to understand. There is already so much abbreviations for other names in the text that the author's message gets lost

Response: Thank you for raising this concern, and we apologize for the lack of clear explanations for these abbreviations. In our manuscript, "P7/6dps" represents postnatal day 7, specifically 6 days post sham operation. "P7/6dpr" represents postnatal day 7, which is 6 days post apical resection performed on postnatal day 1. Similarly, "P56/6dps" and "P56/6dpi" respectively represent heart tissue samples from the control group at postnatal day 56, 6 days post sham operation, and heart tissue samples from the myocardial infarction group at postnatal day 56, 6 days post-myocardial infarction. We have also compiled a separate list of all the abbreviations used in the study to minimize any unnecessary misunderstandings (Page 4-5, line 59-115).

3. why is P7/6dpr and P56/6dpi in Sup. fig1E and P7/6dpr and P56/6dpi in Sup. Fig 1F are being compared where elsewhere the same ages (P7 and P56) are compared. My confusion is again with these nomenclature, it is just too confusing to first figure out what the group is and why particular groups are being compared to

Response: Thank you for your valuable question. In **Supplementary Figure 1E** and **Supplementary Figure 2C**, we evaluated the expression patterns of PMP70 and PEX3 in the injured zone of cardiac tissue in neonatal and adult mice post-injury using immunohistochemistry. The primary purpose was to compare the protein expression differences between the injured and non-injured areas. The terms "P7/6dpr" (6 days post apical resection at postnatal day 1) and "P56/6dpi" (6 days post-myocardial infarction at postnatal day 56) merely indicate the time points at which the assessments were conducted, and they are not intended for direct comparison between these two-time points.

4. supplemental table 1: are these annotations the authors made or based on literature? if based on literature, please cite

Response: Thank you for your thoughtful suggestion. We apologize for the lack of clarity on this matter. Indeed, the descriptions of the biological functions of peroxins family members in **Supplemental Table 1** are derived from previously reported results in the literature. We have now added appropriate citations to the relevant literature in the **Supplementary Materials**.

5. does weak western blot PEX3 bands in Pex3-KO in sup. Fig 2C mean an incomplete knockout?

Response: Thank you for raising this meaningful question. In our study, the homozygous PEX3 knockout mice used were generated with cardiac-specific deletion driven by the Myh6 promoter. We did not intervene in the trace expression of PEX3 in other cell types within the myocardium. Therefore, residual traces of PEX3 bands were observed in the western blot results.

6. can the authors explain why a lower percentage of mono nucleated CMs in Pex3-KO is expected? Isn't the increase in bi-nucleation an indicator of more mature CMs?

Response: Thank you for raising this valuable scientific question. It is well known that adult cardiomyocytes exist in various forms of terminal differentiation, including multinucleation and binucleation, which lack cytoplasmic division and proliferation capacity. In contrast, mononuclear cardiomyocytes represent a relatively immature state of differentiation and possess significant proliferative ability, capable of completing the cell cycle activities, including DNA replication followed by cytokinesis. Therefore, an increase in the proportion of mononuclear cardiomyocytes in cardiac tissue is often considered indicative of activated proliferation and regeneration potential¹⁻⁴. In our study, we found that the proportion of mononuclear cardiomyocytes decreased after cardiomyocyte-specific PEX3 knockout, leading to loss of proliferation and myocardial regeneration capacity, consistent with previous research findings.

1. Bersell K, Arab S, Haring B, Kühn B. Neuregulin1/ErbB4 signaling induces cardiomyocyte proliferation and repair of heart injury. *Cell*. 2009 Jul 23;138(2):257-70. PMID: 19632177.

2. Chakraborty S, Sengupta A, Yutzey KE. Tbx20 promotes cardiomyocyte proliferation and persistence of fetal characteristics in adult mouse hearts. *J Mol Cell Cardiol*. 2013 Sep;62:203-13.

Epub 2013 Jun 7. PMID: 23751911.

3. Becker C, Hesse M. Role of Mononuclear Cardiomyocytes in Cardiac Turnover and Regeneration. *Curr Cardiol Rep.* 2020 May 19;22(6):39. PMID: 32430578; PMCID: PMC7237397.

4. Windmueller R, Leach JP, Babu A, Zhou S, Morley MP, Wakabayashi A, Petrenko NB, Viatour P, Morrisey EE. Direct Comparison of Mononucleated and Binucleated Cardiomyocytes Reveals Molecular Mechanisms Underlying Distinct Proliferative Competencies. *Cell Rep.* 2020 Mar 3;30(9):3105-3116.e4. PMID: 32130910; PMCID: PMC7194103.

7. why was the timepoint P56 not used for Pex3-KO experiments? Inclusion of this timepoint in the analysis for proliferation would help link it to figure 1

Response: Thank you for your suggestion. It is well-known that adult cardiomyocytes have limited proliferative capacity and cannot undergo significant regeneration after injury. Conversely, neonatal mouse hearts have remarkable regenerative abilities and can spontaneously regenerate tissue and restore function after injury, but this capability sharply declines around one week after birth¹⁻². Therefore, by 3-4 weeks after birth, when the mice are around P56, the cardiac tissue has already differentiated and matured, losing its regenerative capacity. This is why we did not perform additional testing at the P56 time point in the PEX3-KO experiments.

1. Cardoso AC, Lam NT, Savla JJ, Nakada Y, Pereira AHM, Elnwasany A, Menendez-Montes I, Ensley EL, Petric UB, Sharma G, Sherry AD, Malloy CR, Khemtong C, Kinter MT, Tan WLW, Anene-Nzulu CG, Foo RS, Nguyen NUN, Li S, Ahmed MS, Elhelaly WM, Abdisalaam S, Asaithamby A, Xing C, Kanchwala M, Vale G, Eckert KM, Mitsche MA, McDonald JG, Hill JA, Huang L, Shaul PW, Szweda LI, Sadek HA. Mitochondrial Substrate Utilisation Regulates Cardiomyocyte Cell Cycle Progression. *Nat Metab.* 2020 Feb;2(2):167-178. Epub 2020 Feb 20. PMID: 32617517; PMCID: PMC7331943.

2. Karra R, Poss KD. Redirecting cardiac growth mechanisms for therapeutic regeneration. *J Clin Invest.* 2017 Feb 1;127(2):427-436. Epub 2017 Feb 1. PMID: 28145902; PMCID: PMC5272171.

8. It is not clear what age the cardiomyocytes are for the PEX3i experiments and how they relate to the KO data

Response: Thank you for your question. In our study, experiments involving PEX3 inhibition (PEX3i) were conducted using primary cardiomyocytes isolated from one-day-old neonatal mice in vitro. We have provided detailed methodologies and experimental strategies in the Methods section. The purpose of conducting PEX3i experiments in vitro was to eliminate the influence of other non-cardiomyocyte cells and to investigate the effect of reducing PEX3, specifically in cardiomyocytes on their proliferative capacity in a controlled in vitro environment. This was complemented by in vivo experiments using cardiomyocyte-specific PEX3 knockout mice to confirm the impact of PEX3 on cardiomyocyte proliferation and myocardial regeneration. The combination of in vitro and in vivo experiments helped elucidate the role of PEX3 in regulating cardiomyocyte cell cycle progression and proliferative activity.

9. im confused in line 206 where there authors mention a knockdown strategy yet the preceding text from line 209 onwards talks about over expression

Response: Thank you for your inquiry. Indeed, the content you mentioned refers to our

investigation of the impact of overexpression or knockdown of cardiomyocyte PEX3 on cardiomyocyte proliferative capacity in primary cardiomyocytes. Our results indicate that overexpression of PEX3 promotes, while knockdown of PEX3 inhibits cardiomyocyte proliferation. Therefore, we focused more on the positive effect of PEX3 overexpression on cardiomyocyte proliferative capacity in **Figure 3**, while the data related to PEX3 knockdown were included in **Supplementary Figure 5**.

10. can the authors provide data that Ad5 transfection is cardiomyocyte specific?

Response: Thank you for raising this meaningful question. In our study, the Ad5 transfection employed was cardiomyocyte-specific. Firstly, all Ad5 vectors utilized cardiac troponin T (cTnT) as the promoter. Additionally, we have provided supplementary data in **Figure 3A** and **Supplementary Figure 5D**. We isolated primary cardiomyocytes and non-cardiomyocytes from neonatal mice and cultured them separately. Then, we transfected both groups of cells with Ad5: cTNT-PEX3, Ad5: cTNT-PEX3i and empty virus, respectively. Through Western blot experiments, we validated the transfection efficiency in cardiomyocytes and non-cardiomyocytes groups. The results showed that transfection with Ad5: cTNT-PEX3 significantly increased, while Ad5: cTNT-PEX3i decreased the protein expression levels of PEX3 in cardiomyocytes and there was no significant change in the non-cardiomyocytes group. Above issue is represented for clarity in the revised manuscript.

11. can the authors provide data that the cardiomyocyte populations used for Ad5 transfection are pure and non-cardiomyocytes obtained from the isolations are not affecting the downstream analysis?

Response: Thank you for your response. Our team has extensive experience in isolating high-purity primary mouse cardiomyocytes, and relevant studies have been published¹⁻³. To validate the purity of the isolated cardiomyocytes in this study, we performed immunofluorescence staining for the cardiomyocyte marker gene cTNT and the nuclear dye Hoechst to calculate the proportion of cardiomyocytes in the total extracted cell population, and the results showed that the proportion of cardiomyocytes and total cells ratio is around 90% (**Supplementary Figure 5A**). In addition, we also proved that the expression of PEX3 in non-cardiomyocytes was not affected by the transfection of Ad5 vectors in **Figure 3A** and **Supplementary Figure 5D**.

1. Fan Y, Cheng Y, Li Y, Chen B, Wang Z, Wei T, Zhang H, Guo Y, Wang Q, Wei Y, Chen F, Sha J, Guo X, Wang L. Phosphoproteomic Analysis of Neonatal Regenerative Myocardium Revealed Important Roles of Checkpoint Kinase 1 via Activating Mammalian Target of Rapamycin C1/Ribosomal Protein S6 Kinase b-1 Pathway. *Circulation*. 2020 May 12;141(19):1554-1569. Epub 2020 Feb 26. PMID: 32098494.

2. Li YF, Wei TW, Fan Y, Shan TK, Sun JT, Chen BR, Wang ZM, Gu LF, Yang TT, Liu L, Du C, Ma Y, Wang H, Sun R, Wei YY, Chen F, Guo XJ, Kong XQ, Wang LS. Serine/Threonine-Protein Kinase 3 Facilitates Myocardial Repair After Cardiac Injury Possibly Through the Glycogen Synthase Kinase-3 β / β -Catenin Pathway. *J Am Heart Assoc*. 2021 Nov 16;10(22):e022802. Epub 2021 Nov 2. PMID: 34726469; PMCID: PMC8751936.

3. Chen BR, Wei TW, Tang CP, Sun JT, Shan TK, Fan Y, Yang TT, Li YF, Ma Y, Wang SB, Wang ZM, Wang H, Shi JZ, Liu L, Chen JW, Zhou LH, Du C, Sun R, Wang QM, Wang LS.

MNK2-eIF4E axis promotes cardiac repair in the infarcted mouse heart by activating cyclin D1. *J Mol Cell Cardiol.* 2022 May;166:91-106. Epub 2022 Feb 27. PMID: 35235835.

12. can the authors provide data that the AAV9 transfections is targeting only cardiomyocytes?

Response: Thank you for your suggestion. In our in vivo experiments, we utilized AAV9 as the vector and cTNT as the promoter to achieve cardiomyocyte-specific intervention with PEX3 overexpression. Since our AAV9:cTNT-PEX3 vector carries an exogenous flag tag sequence, we conducted immunofluorescence staining co-localization experiments for the flag tag and cTNT to validate that our AAV9 transfection specifically targets cardiomyocytes (**Supplementary Figure 7A**).

REVIEWERS' COMMENTS:

Reviewer #1 (Remarks to the Author):

The revised manuscript by Sun et al defines the role of PEX3 during cardiac regeneration. The authors have addressed my comments, and I don't have additional comments.

Reviewer #2 (Remarks to the Author):

I am satisfied with the revised manuscript. The overall paper is very interesting and will contribute to the understanding of heart regeneration.

I have some small minor comments that may need to be fixed but i don't need to see the revised manuscript again:

Line 75: typo? (wild type instead of wildnm type)?

Line 111: typo

Responses to the editorial office and reviewers:

Thanks for your advice and we are delighted to have the opportunity to publish our manuscript entitled "*PEX3 promotes regenerative repair after myocardial injury in mice through facilitating plasma membrane localization of ITGB3*" (Manuscript Number: COMMSBIO-24-0061A) in *Communications Biology*. We are grateful to the editors and reviewers for thoughtful and helpful comments on our manuscript. We carefully revised the manuscript according to the format requirements of *Communications Biology*.

Thanks again for your consideration.

Yours sincerely,
Lian-sheng Wang, PhD, MD.

Reviewer #2 (Remarks to the Author):

I am satisfied with the revised manuscript. The overall paper is very interesting and will contribute to the understanding of heart regeneration.

Response: Thank you for reading our manuscript and giving us some pertinent and constructive comments.

I have some small minor comments that may need to be fixed but i don't need to see the revised manuscript again:

Line 75: typo? (wild type instead of wildnm type)?

Line 111: typo

Response: Thank you for the helpful comments. We have corrected “wildnm type” to “wild type” in our manuscript.